# Can LLMs Solve Molecule Puzzles? A Multimodal Benchmark for Molecular Structure Elucidation

**Kehan Guo**[1]*, **Bozhao Nan**[2]*, **Yujun Zhou**[1], **Taicheng Guo**[1], **Zhichun Guo**[1], **Mihir Surve**[2],
**Zhenwen Liang**[1], **Nitesh V. Chawla**[1], **Olaf Wiest**[2], **Xiangliang Zhang**[1]†
[1]Department of Computer Science and Engineering, University of Notre Dame
[2]Department of Chemistry and Biochemistry, University of Notre Dame
{kguo2,bnan,xzhang33}@nd.edu
https://kehanguo2.github.io/Molpuzzle.io/

## Abstract

Large Language Models (LLMs) have shown significant problem-solving capabilities across predictive and generative tasks in chemistry. However, their proficiency in multi-step chemical reasoning remains underexplored. We introduce a new challenge: molecular structure elucidation, which involves deducing a molecule's structure from various types of spectral data. Solving such a molecular puzzle, akin to solving crossword puzzles, poses reasoning challenges that require integrating clues from diverse sources and engaging in iterative hypothesis testing. To address this challenging problem with LLMs, we present **MolPuzzle**, a benchmark comprising 217 instances of structure elucidation, which feature over 23,000 QA samples presented in a sequential puzzle-solving process, involving three interlinked sub-tasks: molecule understanding, spectrum interpretation, and molecule construction. Our evaluation of 12 LLMs reveals that the best-performing LLM, GPT-4o, performs significantly worse than humans, with only a small portion (1.4%) of its answers exactly matching the ground truth. However, it performs nearly perfectly in the first subtask of molecule understanding, achieving accuracy close to 100%. This discrepancy highlights the potential of developing advanced LLMs with improved chemical reasoning capabilities in the other two sub-tasks. Our MolPuzzle dataset and evaluation code are available at this link.

## 1 Introduction

Artificial intelligence (AI) is revolutionizing the field of chemistry, influencing diverse sectors such as industrial chemical engineering [1, 2], drug discovery [3], and chemistry education [4]. In particular, recent studies have highlighted the success of large language models (LLMs) in addressing predictive challenges in chemistry, including molecular property prediction [5], reaction prediction [6], and experiment automation [7]. These advancements suggest significant potential for AI to enhance efficiency and innovation across these critical areas.

We introduce a new chemical challenge to AI, **molecular structure elucidation**. While this critical task has been explored in other contexts, it remains unexplored for large language models (LLMs),

---

*Both authors contributed equally to this work, supported by the NSF Center for Computer-Assisted Synthesis (C-CAS), https://ccas.nd.edu
†Corresponding author.

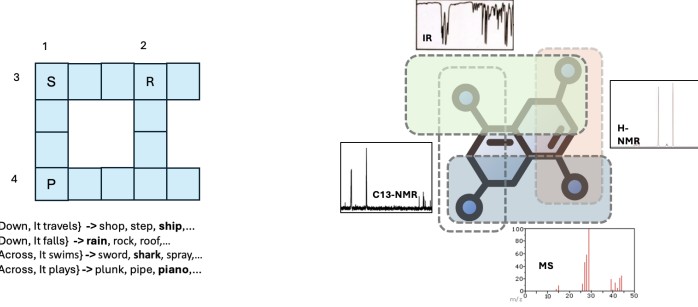

Figure 1: A crossword puzzle (left), and a molecular structure elucidation puzzle (right)

extending beyond familiar predictive and generative domains such as property or reaction prediction, and representing a shift toward complex problem-solving. Analogous to solving a detailed crossword puzzle, **molecular structure elucidation** can be seen as a **molecular puzzle**. It requires the integration of multifaceted data, iterative hypothesis testing, and a deep understanding of chemical cues, much like piecing together clues across a crossword grid to form a coherent solution. Fig. 1 illustrates the problem of molecular structure elucidation alongside its analogical counterpart, the crossword puzzle, highlighting the parallels in strategy and complexity between these two intellectual challenges.

Just as a crossword puzzle requires figuring out words based on given clues and fitting them together in a grid, molecular structure elucidation involves deducing a molecule's structure from various types of data such as nuclear magnetic resonance (NMR), infrared spectroscopy (IR), mass spectrometry, and others. Each type of data provides clues about different aspects of the molecular structure. In a crossword, we integrate clues from across different directions and hints to form words that fit together correctly. Similarly, in molecular structure elucidation, we need to integrate information from different spectroscopic methods to form a consistent picture of the molecule. For example, IR spectra reveal molecular vibrations and functional groups, NMR provides information about the framework of hydrogen and carbon atoms, while mass spectrometry can offer insights into the molecular weight and possible fragmentations.

Nevertheless, molecular structure elucidation is a challenging and time-consuming task. Training undergraduate students in chemistry to solve these puzzles has been a part of the curriculum because determining the structure of molecules is a fundamental skill in the field. Typically, even a single molecule puzzle question on a final exam can take 10 to 15 minutes to solve [8], demanding considerable memory and processing skills from the students. In the domain of complex molecule research, the process of molecular deduction can become even more complex and time-consuming. Therefore, fully automating this process is highly beneficial for accelerating the design of new materials and drugs, as well as enhancing the efficiency of chemical research [9, 10]. However, it remains a challenging task due to the complexities involved in interpreting spectral data and solving intricate reasoning problems associated with molecular structures [11].

In this work, we aim to present molecular structure elucidation in formats that LLMs can effectively process. By adapting this complex task to be compatible with LLMs, we explore their potential as promising tools in chemical research. If successful, LLMs could significantly accelerate scientific discovery in chemistry, transforming how we approach and solve intricate molecular puzzles.

To achieve our objectives, we first introduce a novel dataset named **MolPuzzle**, which includes 234 instances of structure elucidation challenges inspired by common chemistry tasks. Unlike datasets used in predictive or generative tasks, which typically consist of a collection of independent samples and are relatively straightforward to construct, each instance in the MolPuzzle dataset is uniquely complex. It is structured as a sequential process involving three interlinked sub-tasks: **molecule understanding**, **spectrum interpretation**, and **molecule construction**. These instances are accompanied by multimodal data, including images of IR, MASS, H-NMR, and C-NMR spectra, alongside their corresponding molecular formulas. Presenting such a complex, multimodal problem in a format that LLMs can effectively process presents a unique challenge. We, a team of AI researchers

and chemists, are dedicated to formulating the molecule puzzle instances in descriptive languages that are accessible to LLMs. Our focus is on ensuring the utility of these instances, as well as their comprehensive coverage over various scenarios and challenges that mimic real-world conditions. By doing so, **MolPuzzle** opens the door for LLMs to contribute meaningfully to the field of chemistry, potentially accelerating scientific discoveries and innovations.

Second, we present our effort to automate the solving of molecular structure elucidation using LLMs. While certain sub-tasks, such as translating an IR spectrum into a molecular formula, may be solvable by encoder-decoder models [12], the comprehensive resolution of the entire molecular puzzle likely requires the advanced planning and reasoning capabilities of LLMs. We tested 11 state-of-the-art LLMs including GPT-4o, Gemini-pro, and Claude-3-opus. We also conducted a human baseline to compare the performance of humans and LLMs in solving the same puzzles. The **key findings** are: 1) GPT-4o significantly outperforms other LLMs; 2) The best-performing LLM, GPT-4o, performs significantly worse than humans, with only a small portion (1.4%) of its answers exactly matching the ground truth; and 3) GPT-4o's performance primarily collapses in the Stage-2 of spectrum interpretation and gets worse in the Stage-3 of molecule construction, although it performs nearly perfectly in Stage-1 of molecule understanding (with accuracy close to 100%).

To summarize, our key contributions in this work are the presentation of:

- **A new reasoning problem for AI community**. As the focus of AI development has evolved from solving predictive tasks and generative tasks to engaging in complex reasoning tasks—akin to system 2 level thinking—we introduce a reasoning task centered around molecular structure elucidation. This crucial problem from the field of chemistry sets a high benchmark for AI models to reach. Solving this task requires AI models to possess the ability to interpret spectral images, engage in complex reasoning, and plan effectively across extended workflows. This not only challenges the current capabilities of AI but also pushes the boundaries of what AI can achieve in scientific domains, particularly in understanding and manipulating molecular structures.

- **A new light of AI solutions for chemistry community**. By proposing the **MolPuzzle dataset**, we establish another bridge between the fields of AI and chemistry. This initiative leverages the important capabilities of multimodal LLMs, providing the chemistry community with innovative solutions to accelerate the process of structure elucidation. Our initial exploration serves as a demonstration of the potential for these technologies. It sets the stage for further collaborative efforts, inspiring researchers from both domains to collaboratively explore new frontiers in scientific discovery.

The paper is organized as follows. Section 2 presents the related work. In Section 3, we elaborate on the curation of the MolPuzzle dataset. In Section 4, we report the usage of multimodal LLMs in solving MolPuzzle. In Section 5, we discuss the main findings and directions opened by this work. In section 7, we discuss the broader impact of our work. Last, we summarize the study in Section 8 and offer our conclusions.

## 2 Related Work

**Molecular Structure Elucidation.**  Automated molecular structure determination has been researched for decades [13, 14, 15, 16, 17], initially focusing on rule-based systems [18, 19] that interpret spectral data using predefined chemical rules and expert knowledge. Notable examples include SENECA [20], employing genetic algorithms on NMR data, and ACD/Structure Elucidator [21], a commercial software integrating various spectral data. While effective for well-characterized compounds, rule-based methods struggle with complex or novel molecules that deviate from established patterns, and their proprietary nature limits benchmarking accessibility.Machine learning approaches [22, 23, 24, 25, 26, 27, 28, 29] have also been explored. Early studies utilized neural networks to assign infrared spectra to molecular structures [30], and recent advancements leverage deep learning for complex datasets [31]. For example, Alberts et al. [12] used a transformer-based model to predict SMILES strings from IR spectra, later extending this to NMR data analysis [27]. However, most existing research focuses on molecule elucidation using single-type spectrum data, sufficient for simple molecules but inadequate for complex ones since each spectrum provides only

partial structural information. Our study aims to leverage the reasoning and planning capabilities of multimodal large language models (MLLMs) to integrate diverse spectral data, addressing challenges in complex real-world chemistry tasks. We focus on solving the entire puzzle using multiple clues rather than deciphering one word from a single clue.

**Multimodal Benchmarks for LLMs.** With the advancements in developing multimodal LLMs [32, 33, 34, 35, 36], a number of multimodal benchmarks have been curated. These benchmarks are crucial for evaluating and refining the capabilities of MLLMs to process and integrate diverse data types, such as text, images, and audio, for a cohesive understanding. Notably, a benchmark proposed by Yue et al. [37] assesses the reasoning abilities of MLLMs in various college-level subjects. Similarly, MathVista [38] explores MLLMs' multimodal reasoning capabilities in mathematics, while Yin et al. [39] introduced LAMM, a dataset focusing on multimodal instruction tuning and the LabSafetyBench [36] assessed the reliability and safety awareness of LLMs in laboratory environments. Our research shifts the focus to the chemistry domain [6, 40]. To our knowledge, this study is the first to adopt a realistic chemistry task for MLLM processing and to conduct a thorough evaluation of these models' proficiency in chemistry-related reasoning and image analysis. This specialized focus will enhance our understanding of MLLMs' capabilities within a specific scientific domain.

# 3 The MolPuzzle Dataset

Existing benchmarks of chemical tasks primarily focused on predictive or generative tasks involving collections of independent samples that were relatively straightforward to construct. In contrast, our dataset, MolPuzzle, aims to characterize an intertwined assessment of chemistry reasoning and visual understanding, testing the application of AI-assisted technology towards broader scientific discovery. Our data collection process is rigorously designed and implemented by a team uniquely qualified for this task, consisting of esteemed researchers in chemistry and experienced AI specialists who have previously tackled complex chemistry problems. This collaboration ensures that the MolPuzzle dataset not only accurately reflects real-world chemical phenomena and challenges but is also structured in a way that optimally facilitates access and usability for LLMs.

The basic principles guiding our data curation for the MolPuzzle dataset are: 1) ensuring comprehensive coverage by including a wide range of tasks that synthesize visual context with chemical knowledge, facilitating thorough evaluations; 2) varying levels of difficulty to challenge LLMs and highlight their potential limitations; 3) ensuring robust assessment outcomes, i.e., the results are definitive and reliable; and 4) incorporating human expert analysis to identify strengths and weaknesses in model performance, significantly enhancing our understanding of LLMs capabilities.

In this section, we outlined the construction process for the MolPuzzle dataset. We detailed the creation of puzzle tasks in three stages (3.1), as well as the QA pairs involved in these tasks (3.2). Examples are presented in Fig. 2.

## 3.1 Task Construction

Just like a word puzzle where each clue progressively reveals the final answer, the solution to a molecule puzzle is a SMILES string that captures the interconnected substructures of a molecule. We design our molecule puzzles so that solving one requires the accurate identification and integration of each substructural clue, gradually unveiling the complete SMILES representation of the molecule. This approach is inspired by the analytical strategies employed by chemists in the real world, who interpret spectral data and chemical properties to deduce the structures of unknown molecules. Our puzzle-building process mirrors this scientific exploration, arranging clues in a sequence from simple to complex, where each clue builds upon the insights gained from the previous one, requiring precision and careful thought at every stage. We next provide more details on our clue design methodology.

**The Initial Stage (Molecule Understanding).** In designing a molecule puzzle, the first stage involves determining how many building blocks, or substructures, are available. This foundational step is crucial as it sets the stage for constructing the molecule's complete structure, akin to identifying the key pieces in a complex jigsaw puzzle. Starting with the initial hint: A molecular formula, derived from a mass spectrum, indicates the exact types and numbers of atoms in a molecule (e.g., $C15H22O_2$,

representing carbon, hydrogen, and oxygen), chemists can begin to deduce possible structures from the degree of saturation which is calculated based on the number of rings and multiple bonds present in the molecule, the potential for forming aromatic rings, or the presence of functional groups. The initial information provides a preliminary range of building blocks, which can later be selected and assembled to solve the molecular puzzle. To benchmark the capability of LLMs in this stage, we developed 26 unique templates (see Appendix A.2 for details), targeting key analytical tasks such as saturation identification, aromatic ring identification, functional group identification, and saturation degree calculation. This initiative produced 5,859 QA-format pairs, effectively evaluating the models' capacity to understand and process molecular data. Details of these samples are reported in Appendix A.3.

**The Second Stage (Spectrum Interpretation).** With the initial building blocks of the molecule identified from the molecular formula, the next critical step involves refining these components through detailed spectral analysis. Spectrum images such as IR, MASS, $^1$H-NMR, and $^{13}$C-NMR serve as new hints, each adding layers of information akin to clues in a complex puzzle. These spectral images are pivotal in confirming or revising the initial hypotheses about the molecule's structure. For example, IR spectroscopy can verify the presence of specific functional groups, MASS spectrometry can provide the molecular MASS, molecule mass, and fragmentation patterns, and NMR techniques detail the arrangement of hydrogen and carbon within the molecule. By integrating these new hints, researchers can construct a more robust and experimentally accurate model of the molecule. This process not only theoretically validates each building block but also ensures they align perfectly with empirical data, leading to a comprehensive understanding of the molecular structure. Given the importance of spectral images in this analysis, we have developed specialized question templates to evaluate the proficiency of LLMs in interpreting these images. For instance, we created 17 templates for IR and 12 for each of H-NMR, and C-NMR. Each template, such as 'Analyze the IR spectrum' includes specific queries designed to extract detailed insights, such as 'What does the absorption in 3200-3600 suggest?' This structure enables us to format the questions for Visual Question Answering (VQA), facilitating a systematic approach to query handling. Our method has successfully generated a significant repository of VQA format examples, comprising 3,689 for IR and 2,604 for each of MASS, H-NMR, and C-NMR. A detailed analysis of these tasks is available in Appendix A.4.

**The Final Stage (Molecule Construction).** After completing the first two stages, we can assert that we have gathered the necessary building blocks to assemble the molecule. The assembly process will be guided by insights derived from NMR data. Specifically, $^1$H-NMR provides information about the hydrogen environment in the molecule, such as the number of hydrogen atoms, their types (e.g. aromatic), and their connectivity. Meanwhile, C-NMR provides detailed insights into the carbon framework, indicating whether carbon atoms are part of an aromatic ring or not. The assembly of the final molecular structure is an iterative process, during which functional groups are uncovered based on the specific hydrogen and carbon environments. The approach to assembling the final molecular structure is iterative. Starting with initial building blocks selected from the identified fragment pool, LLMs are prompted to select one structure from the pool step by step, based on the NMR guidance, until the maximum number of iterations is reached or the fragment pool is exhausted. This systematic addition ensures that each step in the assembly process not only fits with the previous structure but also aligns perfectly with the latest spectral data, driving us closer to the accurate molecular configuration. We created 27 task templates for each molecule to assess the capability of LLMs in comprehending NMR spectra. These templates include 5 questions about atom numbers and 22 tasks centered on functional groups, generating a total of 6,318 question-answer pairs. We sample both atom-related questions concerning the number of hydrogens and carbons, as well as those targeting functional groups. To reduce bias and ensure more balanced performance, we balance the distribution of labels in the answers—whether indicating the presence or absence of a functional group or specific environment. This ensures a more unbiased evaluation across the sampled tasks.

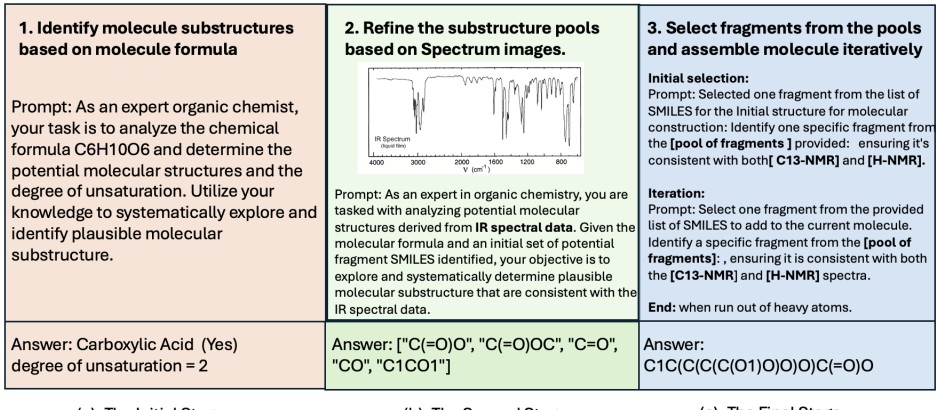

| 1. Identify molecule substructures based on molecule formula | 2. Refine the substructure pools based on Spectrum images. | 3. Select fragments from the pools and assemble molecule iteratively |
|---|---|---|
| Prompt: As an expert organic chemist, your task is to analyze the chemical formula C6H10O6 and determine the potential molecular structures and the degree of unsaturation. Utilize your knowledge to systematically explore and identify plausible molecular substructure. | Prompt: As an expert in organic chemistry, you are tasked with analyzing potential molecular structures derived from **IR spectral data**. Given the molecular formula and an initial set of potential fragment SMILES identified, your objective is to explore and systematically determine plausible molecular substructure that are consistent with the IR spectral data. | **Initial selection:**
Prompt: Selected one fragment from the list of SMILES for the Initial structure for molecular construction: Identify one specific fragment from the **[pool of fragments ]** provided:  ensuring it's consistent with both**[ C13-NMR ]** and **[H-NMR]**.

**Iteration:**
Prompt: Select one fragment from the provided list of SMILES to add to the current molecule. Identify a specific fragment from the **[pool of fragments]**: , ensuring it is consistent with both the **[C13-NMR]** and **[H-NMR]** spectra.

**End:** when run out of heavy atoms. |
| Answer: Carboxylic Acid  (Yes) degree of unsaturation = 2 | Answer: ["C(=O)O", "C(=O)OC", "C=O", "CO", "C1CO1"] | Answer:
C1C(C(C(C(O1)O)O)O)C(=O)O |

|                      |                       |                    |
|----------------------|-----------------------|--------------------|
| (a). The Initial Stage | (b). The Second Stage | (c). The Final Stage |

Figure 2: Examples of QA pairs in the 3 stages of MolPuzzle

## 3.2 QA Sample Derivation

The QA samples for Stage 1 and Stage 2 are automatically generated using their respective question templates (see Appendix A.2) and RDKit [41]. RDKit is an open-source cheminformatics toolkit widely employed for handling chemical informatics data, including molecular structures and fingerprints. This toolkit plays a role in ensuring that the responses, based on the SMILES strings from each molecule puzzle, are accurate and chemically valid. The distribution of these QA samples across different categories is illustrated in Fig. 4. They form a diverse collection of samples for evaluating LLMs' ability to understand molecular formulas and spectra.

The fragment of each QA pair at Stage 3 is initially generated by LLMs, i.e., responding to the prompt 'select one fragment...'. To validate the reliability of these automated generations of QA pairs, experts—two Ph.D. candidates from the chemistry department—manually and independently verified 50 samples, labeling the generated fragments as 'correct' or 'wrong'. Their verification was consistent and demonstrated that 67.4% of examples have correct fragment pools in automated generation. To ensure the quality of derived QA pairs in Stage 3, these chemists manually corrected the fragments pool for each instance in the benchmark.

Fig.3 reports the statistical distribution for the MolPuzzle dataset, which includes 217 puzzle instances (the reasoning of 217 different molecules). Since one puzzle can be solved by different paths, different numbers of QA samples are derived in three stages. We will next evaluate LLMs' performance in solving each puzzle, as well as their capability to solve individual questions.

| Statistic | Number |
|---|---|
| Total MolPuzzle Instances | 217 |
| Stage-1 QA samples | 5,859 |
| - Num. of molecule formula | 176 |
| - Max question length | 128 |
| - Average question length | 94 |
| Stage-2 QA samples | 11,501 |
| - Num. of spectrum images | 868 |
| - Max question length | 340 |
| - Average question length | 264 |
| Stage-3 QA samples | 6,318 |
| - Maximum Iteration | 7 |
| - Max question length | 356 |
| - Average question length | 238 |

Figure 3: Statistic of the MolPuzzle dataset

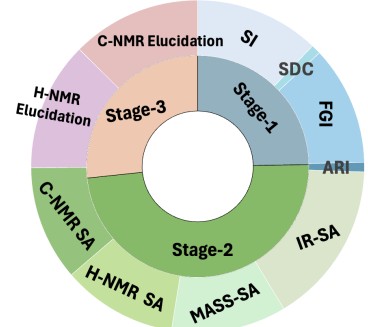

Figure 4: Inner ring: sample distribution in 3 stages. Outer ring: sample distribution across categories in each stage. SI: saturation identification, SDC: saturation degree calculation, FGI: functional group identification, ARI: aromatic ring identification, SA: spectrum analysis.

## 4 Solving MolPuzzle by Multimodal Large Language Models

The reasoning capabilities of foundation models in the chemistry domain remain underexplored. Thus, our aim is to perform both qualitative and quantitative evaluations to systematically assess the reasoning and planning abilities of these models in visual chemistry contexts, using the MolPuzzle benchmark. We first conducted evaluation of a variety of LLMs for completing the individual tasks in each stage, including GPT-4o [42], GPT-3.5-turbo [43], Claude-3-opus [44], Gemini-pro [45], Galactica-30b [46], LLama-3-8B-Instruct [47], Vicuna-13B-v1.5 [48], Mistral-7B-Instruct-v0.3 [49], and in particular multimodal LLMs such as Gemini-pro-vision [45], LLava-Llama-3-8B [50], Qwen-VL-Chat [51], and InstructBlip-Vicuna-7B/13B [32]. Due to space limits, we present only selected results in Table 1 and report the complete list of results in Appendix B. We then assess LLMs' capability to solve the entire puzzles, specifically focusing on how effectively these models can derive the final molecular structure from provided hints (the questions in QA samples). The results are reported in Table 2.

All tasks are evaluated in a zero-shot setting to determine the problem-solving capabilities of LLMs without prior fine-tuning on specific task data. The evaluation process consists of three steps: response generation, answer extraction, and score calculation. More details of the experimental settings including prompts and hyperparameters are presented in Appendix B.1.

To gain an in-depth understanding of the performance of LLMs in comparison with human experts, particularly their failed cases, we invited six Ph.D. candidates in chemistry to solve the puzzles in MolPuzzle, and also assess LLMs' results. More comprehensive details of this **human baseline** and evaluation process are presented in Appendix B.2. The reported performance, including human baselines, is presented as an average with standard deviation over all samples.

### 4.1 LLMs' Performance on Solving Molecule Puzzles

### 4.1.1 Addressing individual QA tasks in three stages

In Table 1, we report the performance of selected LLMs on conducting individual QA tasks in the three stages, including GPT-4o, GPT-3.5-turbo, Claude-3-opus (three top-performing proprietary models), Llama-3-8B-Instruct (the best performing open-source model), and the reference human baseline performance. In stage 2, the variant of Llama3 for a multimodal setting, LLava-Llama-3-8B, is used for handling spectrum image analysis. Since each task involves performing a question-answering task, we evaluate the performance using F1 and accuracy by comparing the LLMs' answers with the ground truth. F1 scores are reported in Table 1, while the accuracy and performance of more LLMs can be found in Appendix B.

The results of Stage 1 (in Table 1 and Appendix Table 3) show that the GPT-4o model excels in these tasks (achieving near-perfect F1 score in 3 out of 4 tasks). The high scores in SI, AI, and FI suggest that LLMs are able to succeed in relatively straightforward chemistry analysis tasks, performing comparably to human experts. However, open-sourced models like LLama3 have limitations in addressing these tasks, possibly due to their limited reasoning abilities in chemistry text-reasoning tasks. In addition, GPT-4o's comparative performance to humans indicates significant advancements in the use of LLMs for complex scientific tasks, suggesting a promising future for leveraging advanced LLMs to improve the efficiency of scientific analysis and discovery.

For the multimodal tasks of Stage 2, GPT-4o remains the top performer, though it exhibits intermediate performance in spectrum interpretation. The F1 scores for the four types of spectra average around 0.6, indicating a moderate level of accuracy in this complex aspect of the challenge. This performance is notably less competitive compared to human baselines, which succeed in approximately 73-77% of the tasks across the four types of spectrum interpretation. This indicates that spectrum interpretation is inherently challenging. While GPT-4o has made significant strides in automated spectrum analysis, there remains considerable room for improvement to bridge the gap between its capabilities and human expertise. More details are presented in Appendix B.4.

The results for Stage 3 indicate that the most advanced LLM, GPT-4o, significantly underperforms compared to the human baseline, with nearly a 40% difference. This might be caused by the fact that

Table 1: F1 scores (↑) of individual QA tasks in three stages. The best LLMs results are in bold font. Tasks in stage 1 are SI-Saturation Identification, ARI-Aromatic Ring Identification, FGI-Functional Group Identification, and SDC-Saturation Degree Calculation.

| Stage 1 (Molecule Understanding) Tasks | | | | |
|---|---|---|---|---|
| Method | SI | ARI | FGI | SDC |
| GPT-4o | **1.00**±**0.000** | 0.943±0.016 | 0.934±0.005 | 0.667±0.003 |
| GPT-3.5-turbo | 0.451±0.025 | 0.816±0.017 | 0.826±0.075 | 0.5±0.099 |
| Claude-3-opus | 0.361±0.009 | **0.988**±**0.015** | **0.934**±**0.001** | **0.856**±**0.016** |
| Galactica-30b | 0.826±0.248 | 0.347±0.000 | 0.467±0.005 | 0.000±0.000 |
| Llama3 | 0.228±0.043 | 0.696±0.051 | 0.521±0.003 | 0.000±0.000 |
| Human | 1.00±0.000 | 1.000±0.000 | 0.890±0.259 | 0.851±0.342 |
| Stage 2 (Spectrum Interpretation) Tasks | | | | |
| Method | IR Interpretation | MASS Interpretation | H-NMR Interpretation | C-NMR Interpretation |
| GPT-4o | **0.656**±**0.052** | **0.609**±**0.042** | **0.618**±**0.026** | **0.639**±**0.010** |
| LLava | 0.256±0.026 | 0.101±0.021 | 0.118±0.008 | 0.254±0.015 |
| Human | 0.753±0.221 | 0.730±0.11 | 0.764±0.169 | 0.769±0.101 |
| Stage-3 (Molecule Construction) Tasks | | | | |
| Method | H-NMR Elucidation | | C-NMR Elucidation | |
| GPT-4o | **0.524**±**0.021** | | **0.506**±**0.037** | |
| Llama3 | 0.341±0.015 | | 0.352±0.017 | |
| Human | 0.867±0.230 | | 0.730±0.220 | |

Table 2: The performance of LLMs and human baseline in solving MolPuzzle. The best LLM results are in bold font. Acc. stands for the Accuracy of Exact Match.

| Method | Acc. (↑) | Levenshtein (↓) | Validity (↑) | MACCS FTS (↑) | RDK FTS (↑) | Morgan FTS (↑) |
|---|---|---|---|---|---|---|
| GPT-4o | **0.014**±**0.004** | **11.653**±**0.013** | **1.000**±**0.000** | **0.431**±**0.009** | **0.293**±**0.013** | 0.232±0.007 |
| Claude-3-opus | 0.013±0.008 | 12.680±0.086 | **1.000**±**0.000** | 0.383±0.050 | 0.264±0.040 | **0.241**±**0.037** |
| Gemini-pro | 0.000±0.000 | 12.711±0.196 | **1.000**±**0.000** | 0.340±0.017 | 0.208±0.002 | 0.171±0.007 |
| Human | 0.667±0.447 | 1.332±2.111 | 1.000±0.000 | 0.985±0.022 | 0.795±0.317 | 0.810±0.135 |

the reasoning ability required for these tasks is complex and multifaceted. When information converges, such as identifying equivalent hydrogen or ring arrangements, a comprehensive understanding of the NMR peaks and their corresponding structures is essential. See more details in Appendix B.5.

### 4.1.2 Addressing entire molecule puzzles

For solving the entire molecule puzzles, the evaluation is limited to the three most advanced multimodal LMMs: GPT-4o [42], Claude-3-opus [44], and Gemini-pro [45], due to the involvement of spectrum image analysis in Stage-2. The results of these models are reported in Table 2, along with those from the human baseline(see complete evaluation process is reported in Appendix C). To comprehensively evaluate the performance, we employ two different types of metrics. The first type of metric measures the chemical similarity between the ground-truth molecules and the generated molecules, assessed using FTS (Fingerprint Tanimoto Similarity) [52] in terms of MACCS [53], RDK [41], and Morgan [54]. Since the generated molecules are in SMILES string format, we also employ natural language processing metrics including the Accuracy of Exact Match [55], and Levenshtein distance [56] (the minimum number of single-character editing required to transform one string into another). Finally, to evaluate whether constructed molecules are valid, we use RDKIT [41] to check the validity of constructed molecules and report the percentage of molecules that are confirmed as valid.

The results in Table 2 show that the best-performed LLM, GPT-4o, is performing much worse than humans, indicating a huge gap between LLMs and humans in solving the molecule puzzles. It is worth noting that all the constructed molecules are valid, even though only a small portion of them (1.4%) exactly match the ground truth. Considering that the accuracy of the exact match is too strict,

we use FTS to analyze more about the chemical closeness of LLMs' answer to the ground truth. A MACCS FTS of 0.431 suggests that the generated molecules maintain a significant level of structural similarity. This indicates that even if the answers are not perfect replicas of the ground truth, they can still be chemically valid and potentially useful as structured hypotheses that could be relived by human scientists.

## 4.2 Success and Failure Analysis

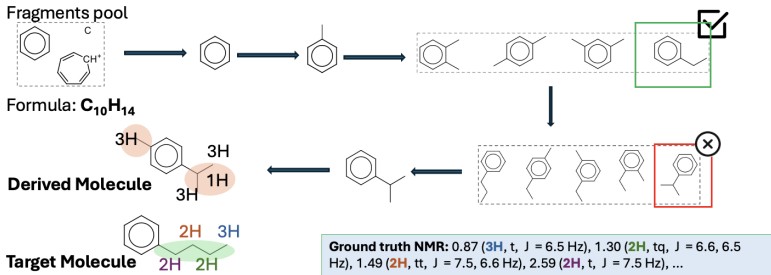

Figure 5: The target molecule contains four distinct non-aromatic hydrogen types, color-coded in the ground truth NMR. However, the model-derived molecule shows hydrogen counts of 3, 3, and 1, differing from the ground truth. The mismatch between the hydrogen types in the green section of the target molecule and the orange region of the predicted molecule results in incorrect fragment selection and assembly.

The above analysis indicates that the most capable model, GPT-4o, performs **nearly perfectly** in Stage-1 of molecule understanding. However, its performance **drops** in Stage-2 for spectrum interpretation, and **worsens further in Stage-3** for molecule construction. We investigate in-depth how GPT-4o eventually fails on most of the puzzles after progressing through the tasks of these three stages. With the help of human evaluators, we gathered all the intermediate steps involved in solving a molecule puzzle and engaged them to scrutinize these steps. Fig. 5 presents case studies that illustrate the iterative steps involved in Stage-3, showcasing the most common errors made by GPT-4o: **the accumulation of errors in iterative steps, which can lead to catastrophic failures**. Note that this stage focuses on selecting the correct fragments and assembling them step by step to form the final molecular structure. We find that GPT-4o can initially succeed in picking the correct fragment when the structure is comparatively simple. However, as the process progresses, it does no select structures that satisfy all the requirements indicated by the NMR data. This difficulty arises because the reasoning requirements expand dramatically as more information and additional constraints need to be incorporated. More qualitative examples can be found in Appendix C.1.

## 5 Findings and Open Directions

Our evaluation has revealed specific limitations of state-of-the-art LLMs in automating molecular structure elucidation. We urge further collaborative efforts from the AI and chemistry communities to design more effective solutions, especially for the tasks in Stage 2 and Stage 3. Based on our findings, we next present the open directions for future research and development.

**Development of Specialized Multimodal LLMs Spectrum Interpretation in Stage 2.** As indicated in our results, the performance of LLMs notably declines beginning in Stage 2, where they struggle with the visual interpretation of 1H and 13C NMR spectra. This difficulty arises because NMR spectra feature sharp, unlabeled peaks with multiplicities that exhibit very small chemical shift differences, making them challenging for visual models to interpret. These multiplicities, however, contain crucial information about the chemical connectivity of molecular fragments. Similarly, closely spaced IR absorptions provide key insights for identifying functional groups. This presents a significant opportunity to develop specialized multimodal LLMs that can more effectively interpret these subtle and complex spectral details.

**Development of New Strategies for Leveraging LLMs in Chemical-Related Planning and Reasoning.** The failure analysis from Stage 3 has motivated us to explore more effective strategies for leveraging LLMs' capabilities in planning and reasoning for fragment selection and assembly. Our first immediate approach was to employ the chain-of-thought technique [57], aiming to provide more structured reasoning and instructions for solving the molecular puzzle. However, despite implementing this method, the results were unsatisfactory, even performing worse than the zero-shot setting we initially reported in the paper. We plan to continue exploring this direction with different implementations and adjustments. A second approach involves utilizing LLMs as agents in a more dynamic and interactive manner. This strategy incorporates feedback loops, allowing the models to iteratively refine their responses based on new information or corrections. By doing so, we aim to mitigate the accumulation of errors in iterative steps and reduce the risk of catastrophic failures during the problem-solving process. In addition, we are investigating fine-tuning strategies to enhance the model's ability to handle domain-specific tasks. This involves fine-tuning LLMs on curated chemical datasets that include detailed annotations of spectral data and molecular structures. The goal is to train the model to recognize subtle patterns and dependencies that are often missed in a general-purpose pre-trained model. By tailoring the model's training to this domain, we expect to improve its reasoning and planning capabilities when interpreting complex spectra and assembling molecular fragments.

## 6   Negative Societal Impacts

Automating molecular elucidation using LLMs has significant benefits but also poses serious risks, especially regarding the creation of prohibited drugs. 1.)Facilitation of Illicit Drug Synthesis: LLMs could be used to design new synthetic drugs that evade current regulations, making it easier for illicit manufacturers to produce harmful substances. 2.)Lowering the Barrier to Entry: The technology could enable individuals with minimal expertise to create detailed molecular blueprints for prohibited drugs, increasing the potential for misuse. 3.) Regulatory Challenges: The rapid generation of novel compounds could overwhelm drug regulators, leading to delays in banning new synthetic drugs and complicating the control of harmful substances. 4.) Ethical and Legal Issues: Questions about responsibility and access to such powerful tools arise. Regulating who can use these technologies and for what purposes becomes crucial to prevent misuse.

## 7   Broader Impact

Our work has broad impacts across multiple dimensions. First, it offers valuable insights and recommendations for both AI researchers and chemists in academia and industry. These perspectives enhance the effective utilization of LLMs and guide future advancements in the field. Second, our approach to benchmarking and improving LLMs through real-world tasks like the MolPuzzle can also foster greater collaboration between computational scientists and chemists. By aligning AI technologies with traditional chemical research, these interdisciplinary efforts can accelerate the discovery of new materials, drugs, and chemical processes, potentially leading to significant advancements in healthcare and industry.

## 8   Conclusion

In this paper, we introduced MolPuzzle, a new benchmark challenge to advance our capabilities in molecular structure elucidation. We evaluated state-of-the-art LLMs on this task, revealing their strengths and limitations in handling complex chemical reasoning. Our analysis highlights significant performance gaps, particularly in spectrum interpretation and molecule construction. These findings not only suggest ways to improve LLM performance but also set the stage for transforming approaches to chemical research. MolPuzzle serves as a critical step toward harnessing the potential of LLMs in chemistry, fostering innovation and collaboration within the AI and chemistry communities to enhance scientific inquiry and application.

## Acknowledgments and Disclosure of Funding

This work was supported by the National Science Foundation (CHE–2202693) through the NSF Center for Computer-Assisted Synthesis (C-CAS).

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

# A MolPuzzle Benchmark Details

This section complements Section 3 with a fine-grained summary of the dataset collection, results validation, and evaluation procedure, along with a fuller characterization of the task instances and the corresponding prompts.

## A.1 Data Collection

The initial molecules were selected by referencing the textbook *Organic Structures from Spectra, 4th Edition*, available as an online PDF on ResearchGate. We chose 234 molecules based on spectrum tasks involving IR, MS, $^1$H-NMR, and $^{13}$C-NMR to reflect a difficulty level suitable for graduate students[8].

To address copyright concerns, we excluded molecules with publicly available mass spectrometry (MS) spectra in open-source databases from our study. The remaining spectra were sourced from public resources, notably the PubChem database[58]. For additional spectra that were not available, we used simulation methods[59][29] and provided a Jupyter notebook to generate these data, ensuring high-quality spectra for analysis. Our final dataset comprised 200 molecules.

Given the challenges associated with NMR spectrum images, some spectra were obtained from simulated data in text format for $^1$H-NMR and $^{13}$C-NMR. This approach ensured clarity and accuracy in the evaluation of molecular structures.

To assess the multiple-stage abilities of LLMs, we designed a unique question-and-answer evaluation. This framework tested the LLMs' capabilities in interpreting and integrating data from different types of spectra, simulating real-world challenges. Details of this evaluation framework are provided in the next section.

## A.2 Template design

Each template was crafted to target specific skills within molecular understanding. For instance, saturation identification challenges the models' ability to discern the degree of saturation in a molecule, which is crucial for understanding its chemical reactivity and stability. Aromatic ring identification tests the models' ability to recognize benzene-like structures, which are fundamental in organic chemistry due to their common occurrence and unique properties. Saturation degree calculation pushes the models to apply quantitative analysis, requiring not just recognition but also computation based on molecular structures.

By diving deeper into the rationale behind each template and the kind of chemical knowledge they are designed to test, we can better appreciate how these tasks simulate real-world applications in chemistry. This approach not only tests the models' basic recognition abilities but also their capacity to perform complex reasoning and apply theoretical knowledge practically. The template examples are in A.3.

## A.3  Stage1 QA Samples

Table 3: QA samples for the molecule understanding task

| Task | Prompt |
|------|--------|
| Saturation Identification | Question: Could the molecule with the formula C8H10O potentially be Saturated?

Answer: No
Model response: No. |
| Aromatic Ring Identification | Question: Could the molecule with the formula C8H10O have aromatic rings?

Answer: Yes
Model response: Yes. |
| Functional Group Identification | Question:Could the molecule with the formula C6H14O2 potentially contain a Amine group, given the Degree of Unsaturation is 0.0?

Answer: No
Model response: No, the molecule doesn't contain Amine group |
| Saturation Degree Calculation | Question: Calculate the Degree of Unsaturation of the molecule with the formula C8H10O?

Answer: 4.0
Model response: 2 |

## A.4  Stage2 QA Samples

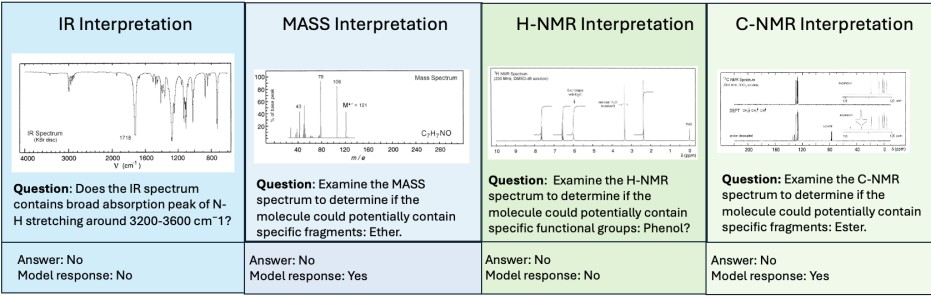

## A.5  Stage3 QA Samples

Table 4: QA samples for the molecule construction task

| Task | Prompt |
|------|--------|
| H-NMR Elucidation | Question: Calculate the number of different types of hydrogen atoms present in the molecule, based on the given H-NMR: 4.51-4.61 (4H, 4.56 (s), 4.56 (s)), 7.06-7.32 (10H, 7.13 (dddd, J = 7.9, 7.7, 1.8, 0.6 Hz), 7.13 (dddd, J = 7.9, 7.7, 1.8, 0.6 Hz), 7.25 (dddd, J = 7.9, 1.5, 1.3, 0.6 Hz), 7.25 (dddd, J = 7.9, 1.5, 1.3, 0.6 Hz), 7.26 (tt, J = 7.7, 1.5 Hz), 7.26 (tt, J = 7.7, 1.5 Hz))

Answer: 4
Model response: 3. |
| C-NMR Elucidation | Question: Analyze the given C-NMR data and determine the number of different types of carbon atoms present in the molecule, based on given C-NMR: 39.3 (1C, s), 63.4 (1C, s), 127.8 (1C, s), 128.4 (2C, s), 128.8 (2C, s), 134.2 (1C, s). Only output the number.

Answer: 6
Model response: 8 |

# B  Evaluation Experiments

## B.1  Experimental Setting

During our testing phase, we selected 100 questions and employed two distinct prompting strategies with the large language model (LLM). Initially, the LLM was tasked with directly answering the

questions. In a subsequent approach, the same queries were presented, but the model was prompted to execute a chain-of-thought reasoning process before responding. Each question in our dataset begins with a comprehensive description of the chemical context, along with specified answer formats and detailed guiding rules. To ensure a balanced representation of each task category, for tasks in Stage 1, the distribution ratio for Saturation Identification (SI), Functional Group Identification (FI), Aromatic Ring Identification (AI), and Saturation Degree Calculation (SC) is set at 2:3:3:2. In Stage 2, we have randomly selected 100 questions from each category of the spectrum. For Stage 3, we randomly selected 100 questions focused on H-NMR and C-NMR analyses.

We carried out this evaluation over three rounds, analyzing responses using both accuracy and the F1 score for tasks involving Saturation Identification (SI), Functional Group Identification (FI), and Aromatic Ring Identification (AI). For Saturation Degree Calculation (SDC), which yields numerical results, we assessed accuracy by comparing the count of correct matches to the ground truth data. The detailed results are reported in Table A.3. To ensure that all results are presented in a way that facilitates direct comparison, only those using similar evaluation metrics(AI, FI, AI) are included in the main table. For the SI, AI, and FI tasks, we use the F1 score and Accuracy to evaluate their performance since they are classification tasks. For the SDC task, the answer is a numerical number, so we only use the accuracy score to measure the performance of the LLMs. This approach helps to keep the evaluation coherent and focused on comparable data points.

## B.2 Human Evaluation

To evaluate the performance of large language models (LLMs) on specialized tasks against expert humans, we recruited six graduate students from chemistry department to solve the MolPuzzle benchmark. These students, having recently completed a graduate-level course in Molecular Structural Elucidation, represented a highly skilled group of human participants.

For the experiment, we randomly selected six questions from the MolPuzzle dataset for each stage of the study. These questions were presented to the students in different formats according to the stage: In Stages 1 and 2, the questions were simple Yes/No or required short answers. In Stage 3, to align with the conventional methods chemists use to express chemical structures, students were asked to upload images of their hand-drawn structures instead of using SMILES strings. These images were manually compared to the ground truth to calculate scores.

We also imposed self-regulated time constraints to mirror the challenging nature of molecular structural elucidation. Beyond individual stage evaluations, we presented each participant with a complete molecule puzzle, consisting of a formula and four spectral images. The students were tasked with solving these puzzles within a 20-minute time frame. Impressively, all participants successfully submitted their solutions within the allotted period.

Our study included a component where human evaluators were involved to assess the performance of the AI models. To ensure the protection and ethical treatment of all participants, we conducted a thorough risk assessment. Potential risks identified included privacy concerns and the mental strain of repetitive tasks. Mitigation strategies, such as ensuring anonymity and providing breaks, were implemented to protect our evaluators.

The study was submitted for review and received approval from our Institutional Review Board (IRB). The IRB approval number is [insert approval number], which verifies that our protocols met all ethical guidelines for research involving human subjects. Throughout the project, we adhered strictly to these protocols to ensure ongoing compliance with ethical standards.

## B.3 Stage1

Molecule understanding requires comprehensive analysis and interpretation of molecular structures, with a focus on chemical properties and spectroscopic data. In our study, we created a dataset of 234 molecules and developed eight distinct question templates across four categories: **Saturation Identification(SI), Functional Group Identification(FI), Aromatic Ring Identification(AI), and**

649 **Saturation Degree Calculation(SC).** These templates assess the ability to identify substructures,
650 compute saturation levels, and infer structural presence, incorporating concepts in the chemistry
651 reasoning process. Each question also necessitates a deep understanding of molecular bonding,
652 stereochemistry, and functional group identification. Responses were generated using the RDKit
653 library, ensuring precise and reliable answers grounded in established chemical informatics.

Table 3: The accuracy(↑), F1 score(↑)in 4 different molecule understanding categories, the best LLMs are in bold font.

| Model | CoT | SI | | AI | | FI | | SC |
|---|---|---|---|---|---|---|---|---|
| | | F1 | Acc | F1 | Acc | F1 | Acc | Acc |
| GPT-4o | - | **1±0.0** | **1±0.0** | 0.943±0.016 | 0.944±0.015 | 0.934±0.005 | 0.966±0.0 | 0.667±0.003 |
| GPT-4o | ✓ | **1±0.0** | **1±0.0** | **0.911±0.031** | **0.911±0.031** | 0.689±0.025 | 0.766±0.027 | 0.816±0.062 |
| GPT-3.5 | - | 0.451±0.025 | 0.825±0.075 | 0.816±0.017 | 0.816±0.075 | 0.826±0.075 | 0.683±0.016 | 0.5±0.099 |
| GPT-3.5 | ✓ | 0.448±0.026 | 0.816±0.008 | 0.798±0.025 | 0.800±0.027 | 0.526±0.053 | 0.622±0.031 | 0.533±0.131 |
| Claude-3-opus | - | 0.361±0.009 | 0.556±0.023 | **0.988±0.015** | **0.988±0.015** | **0.934±0.001** | 0.966±0.001 | **0.856±0.016** |
| Claude-3 | ✓ | 0.760±0.189 | 0.903±0.046 | 0.878±0.025 | 0.867±0.001 | 0.547±0.112 | 0.843±0.081 | 0.900±0.025 |
| Gemini-pro | - | 0.285±0.020 | 0.399±0.040 | 0.775±0.093 | 0.788±0.083 | 0.646±0.052 | 0.748±0.051 | 0.200±0.004 |
| Gemini-pro | ✓ | 0.391±0.045 | 0.651±0.108 | 0.685±0.088 | 0.688±0.087 | 0.562±0.018 | 0.629±0.023 | 0.283±0.062 |
| LLama3 | - | 0.367±0.018 | 0.583±0.047 | 0.490±0.030 | 0.533±0.027 | 0.472±0.133 | 0.588±0.0 | 0.0±0.0 |
| LLama3 | ✓ | 0.473±0.011 | 0.899±0.040 | 0.384±0.026 | 0.533±0.0 | 0.570±0.035 | 0.799±0.047 | 0.017±0.001 |
| Vicuna-13b | - | 0.031±0.022 | 0.033±0.025 | 0.500±0.087 | 0.522±0.083 | 0.308±0.038 | 0.311±0.041 | 0.0±0.0 |
| Vicuna-13b | ✓ | 0.380±0.023 | 0.616±0.062 | 0.342±0.006 | 0.522±0.157 | 0.516±0.080 | 0.855±0.016 | 0.0±0.0 |
| Mistral-7b | - | 0.221±0.014 | 0.283±0.025 | 0.384±0.005 | 0.500±0.0 | 0.319±0.014 | 0.322±0.157 | 0.0±0.0 |
| Mistral-7b | ✓ | 0.433±0.007 | 0.766±0.023 | 0.342±0.006 | 0.522±0.016 | 0.601±0.102 | 0.877±0.031 | 0.0±0.0 |

## B.4 Stage2

655 The Spectrum interpretation tasks mainly measure the capability of LLMs in analyzing images
656 related to identifying key substructures indicated by the spectrum plot. In this study, we utilize
657 four distinct types of spectral images: nuclear magnetic resonance (NMR), infrared spectroscopy
658 (IR), mass spectrometry, and others. Each type of data offers insights into various aspects of the
659 molecular structure. We've created specific question templates for each spectrum, targeting peak
660 and substructure identification factors. These templates are designed manually and emphasize the
661 intricate connection between the spikes or troughs in the figures and the structures of the molecules.
662 Responses were generated using the RDKit library to ensure correctness.

663 The findings from Stage 2 are presented in Table 4. We exclusively focus on the zero-shot learning
664 outcomes, as our observations indicate that implementing chain-of-thought prompting leads to a
665 deterioration in model performance. To address this limitation, we offer qualitative insights in C.1.

Table 4: The accuracy(↑), F1 score(↑) for IR, MASS spectrum, H-NMR, and C-NMR interpretation tasks."-" means the results are not interoperable

| Model | Stage-2 Tasks | | | | | | | |
|---|---|---|---|---|---|---|---|---|
| | IR Interpretation | | MASS Interpretation | | H-NMR Interpretation | | C-NMR Interpretation | |
| | F1 | Acc | F1 | Acc | F1 | Acc | F1 | Acc |
| GPT-4o | **0.656±0.052** | **0.713±0.06** | **0.609±0.042** | **0.767±0.042** | **0.618±0.026** | **0.864±0.007** | **0.639±0.107** | **0.892±0.049** |
| Claude-3-opus | 0.440±0.006 | 0.476±0.055 | 0.398±0.032 | 0.466±0.019 | 0.572±0.190 | 0.842±0.017 | 0.554±0.075 | 0.716±0.042 |
| Gemini-3-pro-vision | 0.194±0.002 | 0.119±0.016 | 0.116±0.036 | 0.124±0.038 | 0.545±0.048 | 0.851±0.062 | 0.492±0.016 | 0.619±0.044 |
| LLava1.5-8b | 0.256±0.026 | 0.414±0.044 | 0.101±0.021 | 0.104±0.26 | 0.118±0.008 | 0.186±0.011 | 0.254±0.015 | 0.472±0.023 |
| Qwen-VL-Chat | 0.243±0.027 | 0.392±0.043 | 0.125±0.006 | 0.116±0.021 | 0.255±0.007 | 0.611±0.031 | - | - |
| InstructBLIP-7b | 0.239±0.020 | 0.263±0.014 | 0.101±0.021 | 0.104±0.26 | - | - | 0.044±0.006 | 0.064±0.023 |
| InstructBLIP-13b | 0.239±0.020 | 0.263±0.014 | 0.101±0.021 | 0.104±0.26 | - | - | 0.047±0.014 | 0.067±0.025 |

## B.5 Stage-3

667 Constructing a molecule involves a detailed analysis of NMR data, which is critical for understanding
668 its structure. H-NMR data are essential as they provide information about the hydrogen environments
669 within the molecule, including the number and types of hydrogen atoms (such as aliphatic or

aromatic), as well as their connectivity. Conversely, C-NMR data offer in-depth insights into the carbon framework, illustrating the distribution and linkage of carbon atoms within the molecule. In our study, to evaluate the ability of large language models (LLMs) to interpret NMR data, we generated 1,171 question-and-answer (QA) pairs. These pairs focus on key NMR interpretation tasks, such as counting hydrogen atom types and identifying substructures, which are critical for accurate analysis.

Despite observing moderate accuracy from the LLMs in Stage 2 of our testing, we enhanced the quality of the QA pairs in Stage 3 by providing the LLMs with verified NMR data, generated by using nmrdb[60]. This approach ensures that the data used is reliable and helps maintain the integrity of our results. The findings from Stage 2 are presented in Table. We exclusively focus on the zero-shot learning outcomes, as our observations indicate that implementing chain-of-thought prompting leads to a deterioration in model performance. To address this limitation, we offer qualitative insights in

Table 5: The F1 score(↑) for H-NMR, and C-NMR Structure Elucidation

| Method | H-NMR Elucidation | C-NMR Elucidation |
|---|---|---|
| GPT-4o | **0.524±0.021** | **0.506±0.037** |
| Claude-3-opus | 0.395±0.008 | 0.313±0.029 |
| Gemini-pro | 0.333±0.012 | 0.308±0.031 |
| Llama3 | 0.341±0.015 | 0.352±0.017 |
| Vicuna-13b | 0.181±0.013 | 0.244±0.001 |
| Mistral-7b | 0.131±0.032 | 0.122±0.027 |

 # C  Complete Molecule Elucidation Process

---

**Algorithm 1** Fragment-Based Molecule Assembly Algorithm

---

> **Input:** Fragment pool (SMILES strings), NMR description, Original molecular formula, Original unsaturation degree
> **Output:** Assembled molecule that satisfies molecular formula and NMR data

---

1: **Initialize:**
2:     Set iteration count $k \leftarrow 0$
3:     Set remaining formula $\leftarrow$ Original molecular formula
4:     Set remaining unsaturation $\leftarrow$ Original unsaturation degree

5: **1. Initial Fragment Selection:**
6: Prompt LLM with fragment pool and NMR description to select an initial fragment
7: Extract and store the selected fragment

8: **2. Chemical Formula and Unsaturation Check:**
9: Convert selected fragment to its chemical formula and unsaturation degree
10: Update remaining formula and unsaturation by subtraction

11: **while** remaining formula has multiple main atoms **and** $k < 5$ **do**
12:     Increment iteration count $k \leftarrow k + 1$

13:     **3. Iterative Fragment Assembly:**
14:     Prompt LLM to select additional fragments considering remaining formula and unsaturation
15:     Concatenate selected fragments to form a potential molecule

16:     **4. Molecule Validation and NMR Matching:**
17:     Validate the new molecule using RDKit for connectivity
18:     **if** multiple valid molecules exist **then**
19:         LLM ranks molecules based on NMR match
20:         Select the molecule that best matches the NMR data
21:     **end if**

22:     **5. Subsequent Assembly and Adjustment:**
23:     After successful connection, update remaining formula and remaining unsaturation
24: **end while**

25: **6. Termination Conditions:**
26: **if** no valid fragments can be selected **or** remaining formula is fully satisfied **or** $k \geq 5$ **then**
27:     Terminate the assembly process
28: **end if**

29: **7. Final Output:**
30: Record the final assembled molecule and intermediate stages
31: **if** final molecule fits original molecular formula and NMR data **then**
32:     **Return** valid solution
33: **else**
34:     **Return** no valid solution found
35: **end if**

---

 ## C.1  Qualitative Results

 In this section, we present several examples using GPT-4's chain-of-thought (CoT) reasoning to
 facilitate a clearer understanding of the results. We have enlisted two Ph.D. candidates from the
 chemistry department to evaluate these CoT outcomes. The analysis uses color coding to indicate the

### C.1.1   Stage 2 examples

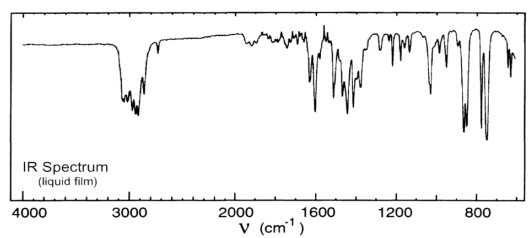

**CoT GPT-4o**
1. **Identify the OH Stretching Vibration:**
• Phenols typically exhibit a broad O-H stretching absorption band in the region of 3200-3600 cm⁻¹.
• In the provided IR spectrum, there is a broad absorption band centered around 3400 cm⁻¹, which is characteristic of an O-H stretch, indicative of a hydroxyl group.
2. **Identify the C-H Stretching Vibration:**
• Aromatic C-H stretching vibrations usually appear in the region of 3000-3100 cm⁻¹.
• The IR spectrum shows sharp peaks in this region, which are consistent with the presence of aromatic C-H bonds.
3. **Identify the C=C Stretching Vibration:**
• Aromatic C=C stretching vibrations appear in the region of 1400-1600 cm⁻¹.
• The spectrum shows significant absorption bands around 1500-1600 cm⁻¹, suggesting the presence of an aromatic ring.
4. **Identify the C-O Stretching Vibration:**
• For phenols, the C-O stretching vibration typically appears in the range of 1200-1300 cm⁻¹.
• The spectrum shows absorptions in this region, which could correspond to the C-O stretch of a phenol.
5. **Examine for Additional Characteristic Absorptions:**
• Phenols may also show additional bands due to aromatic ring vibrations in the region of 600-900 cm⁻¹.
• The spectrum shows multiple absorptions in this region, further supporting the presence of an aromatic ring.

**Question:**

Examine the IR spectrum to determine if the molecule could potentially contain specific functional groups: Phenol. Look for the presence of characteristic absorption bands and analyze the wavenumbers and intensities of these peaks. This analysis will help identify the functional groups and key structural features within the molecule. You should think step by step.

Figure 6: Human annotated IR spectrum interpretation

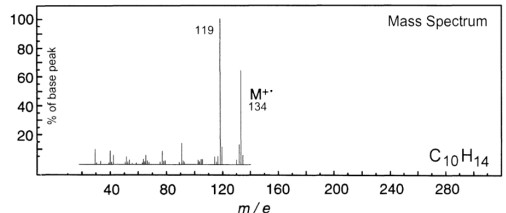

**CoT GPT-4o:** 1.Identify the Molecular Ion Peak (M⁺•):
• The molecular ion peak (M⁺•) is observed at m/z = 134 . This peak represents the molecular weight of the compound.
2. Identify Significant Fragment Peaks:
• The base peak (the most intense peak) is at m/z = 119 . This fragment is often very informative because it is the most stable fragment.
• Other significant peaks are observed at various m/z values (e.g., around 40, 60, 80, etc.).
3. Determine the Mass Difference Between Key Peaks:
• The difference between the molecular ion peak (134) and the base peak (119) is 15. This suggests a loss of a CH₃ (methyl) group, which is common in mass spectrometry fragmentation patterns.
4.Identify Common Ether Fragmentation Patterns:
• Ethers typically show fragmentation patterns where the C-O bond breaks, leading to fragments that indicate the presence of an ether.
• A common ether fragmentation involves the loss of an alkyl group adjacent to the oxygen, often observed as a mass loss of 15 (CH₃), 29 (C₂H₅), or 43 (C₃H₇).
5. Analyze the Presence of Ether Characteristic Peaks:
• Given the mass spectrum data, the loss of 15 units (m/z = 134 to m/z = 119) could suggest the presence of a methyl group, potentially adjacent to an oxygen atom if an ether is present.

**Question:**
Examine the MASS spectrum to determine if the molecule could potentially contain specific fragments: Ether. Look into the number of fragments observed and analyze the differences between the larger fragments. This analysis will help identify the presence of key structural features within the molecule? You should think step by step

Figure 7: Human annotated MASS spectrum interpretation

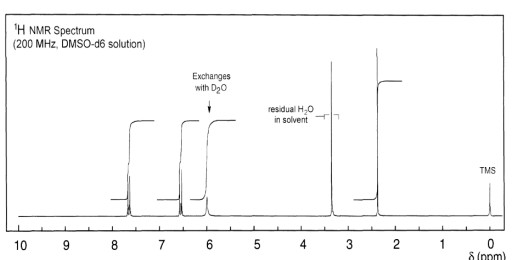

**CoT GPT-4o**: Analyzing the provided ^1H NMR spectrum gives us various insights into the molecular structure of the compound in question. Here's a step-by-step analysis:

**Chemical Shifts and Splitting Patterns**
The spectrum shows peaks at different chemical shift regions, which can be indicative of different types of hydrogen environments:
• **Around 9.5 ppm**: This is a typical region for aldehyde protons (CHO). The singlet peak here suggests the presence of an aldehyde group.
• **7.0 to 8.0 ppm**: These peaks typically indicate aromatic protons. The multiplicity of these peaks suggests that there may be multiple aromatic rings or substituted aromatic rings.
• **5.5 to 6.5 ppm**: Peaks in this region are often associated with hydrogen atoms attached to carbon atoms that are double bonded to other carbon atoms (e.g., vinyl protons) or to heteroatoms.
• **Around 1.5 ppm**: This region is characteristic of methyl groups (CH₃) adjacent to heteroatoms or double bonds.

**Integration and Relative Intensity**
• The spectrum does not provide integration values, but typically, the area under each peak corresponds to the number of equivalent protons contributing to that signal.

**Identification of Functional Groups**
• **Aldehyde Group**: The presence of a sharp singlet peak at around 9.5 ppm strongly suggests an aldehyde functional group.
• **Aromatic Rings**: The cluster of peaks from 7.0 to 8.0 ppm suggests the presence of aromatic protons. The nature of splitting could indicate substitution patterns on the aromatic ring.
• **Vinyl or Alkene Groups**: Peaks in the 5.5 to 6.5 ppm range could indicate the presence of vinyl protons, suggesting a double bond adjacent to the aromatic ring or another functional group.
• **Methyl Groups**: The peak at about 1.5 ppm suggests the presence of methyl groups, possibly adjacent to a heteroatom or a double bond, influencing its chemical shift.

**Question:**
Examine the H-NMR spectrum to determine if the molecule could potentially contain specific functional groups: Aldehyde? Look for the presence of characteristic absorption bands and analyze the wavenumbers and intensities of these peaks. This analysis will help identify the functional groups and key structural features within the molecule. You should think step by step

Figure 8: Human annotated H-NMR spectrum interpretation

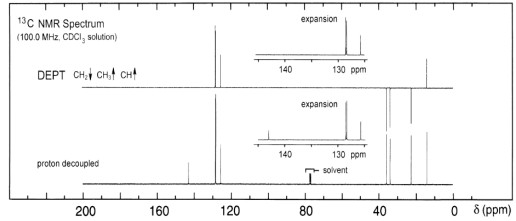

To determine if the molecule could potentially contain an alcohol group, we need to analyze the ^{13}C NMR spectrum step by step.

Step-by-Step Analysis of the ^{13}C NMR Spectrum:

1.   Identify the Chemical Shifts:
•     ^{13}C NMR chemical shifts can help identify the type of carbon atoms present in the molecule. Alcohol carbons typically appear in specific regions:
•     Carbons attached to hydroxyl groups (C-OH) typically resonate between 50-80 ppm.
2.   Examine the DEPT Spectrum:
•     DEPT (Distortionless Enhancement by Polarization Transfer) spectra differentiate between CH, CH₂, and CH₃ groups.
•     CH and CH₃ groups appear as positive peaks (upward), while CH₂ groups appear as negative peaks (downward).
3.   Identify Key Peaks in the Spectrum:
•     Look for peaks in the region of 50-80 ppm, which could indicate the presence of carbons bonded to an oxygen atom (C-OH).
•     Compare the DEPT spectrum to identify whether these peaks correspond to CH (upward peaks) or CH₂ (downward peaks).

Detailed Analysis:

1.   Chemical Shifts in the Range of 50-80 ppm:
•     The spectrum shows peaks in the region of 50-80 ppm. These peaks could correspond to carbons bonded to an oxygen atom, indicating a potential alcohol group.

**Question:**
Examine the C-NMR spectrum to determine if the molecule could potentially contain specific fragments: Alcohol. Look into the number of fragments observed and analyze the differences between the larger fragments. This analysis will help identify the presence of key structural features within the molecule?

Figure 9: Human annotated C-NMR spectrum interpretation

## C.1.2 Stage 3 examples

As an expert organic chemist, your task is to analyze and determine the potential structures that can be derived from a given NMR data. Analyze the given C_NMR data and determine the number of different types of carbon atoms present in the molecule based on given C_NMR: ['δ 39.3 (1C, s), 63.4 (1C, s), 127.8 (1C, s), 128.4 (2C, s), 128.8 (2C, s), 134.2 (1C, s)']. You should think step-by-step.

**CoT GPT-4o**

1. **Identify the Chemical Shifts and Corresponding Carbon Types:**
• δ 39.3 (1C, s): This chemical shift is typically associated with aliphatic carbons, such as those in a methyl or methylene group adjacent to electronegative atoms.
• δ 63.4 (1C, s): This chemical shift is usually indicative of a carbon atom bonded to an oxygen atom, such as in an alcohol or ether group.
• δ 127.8 (1C, s): This shift falls within the region typical for aromatic carbons.
• δ 128.4 (2C, s): These shifts are also within the aromatic region and suggest two equivalent aromatic carbon atoms.
• δ 128.8 (2C, s): Similarly, these shifts suggest two more equivalent aromatic carbon atoms.
• δ 134.2 (1C, s): This shift is also indicative of an aromatic carbon, possibly a substituted one or part of a double bond adjacent to an aromatic ring.
2. **Count the Number of Distinct Carbon Environments:**
• **Aliphatic Carbon:** 1 carbon at δ 39.3
• **Carbon-Oxygen (Alcohol/Ether) Carbon:** 1 carbon at δ 63.4
• **Aromatic Carbons:** 1 carbon at δ 127.8, 2 equivalent carbons at δ 128.4, 2 equivalent carbons at δ 128.8, 1 carbon at δ 134.2
3. **Summarize the Types of Carbon Atoms:**
• **Aliphatic carbon (single):** 1 type (1 carbon)
• **Carbon bonded to oxygen:** 1 type (1 carbon)
• **Aromatic carbons:**
• 1 type at δ 127.8 (1 carbon)
• 1 type at δ 128.4 (2 equivalent carbons)
• 1 type at δ 128.8 (2 equivalent carbons)
• 1 type at δ 134.2 (1 carbon)

Figure 10: Human annotated H-NMR Elucidation

As an expert organic chemist, your task is to analyze and determine the potential structures that can be derived from a given NMR data. Generate the number of different types of hydrogen atoms present in the molecule based on given H_NMR: ['δ 2.23 (3H, s), 3.79 (3H, s), 7.04 (2H, ddd, J = 8.3, 1.3, 0.4 Hz), 7.97 (2H, ddd, J = 8.3, 1.6, 0.4 Hz)']. You should think step by step.

**CoT GPT-4o**

1. **Identify the Chemical Shifts and Corresponding Hydrogen Types:**
• δ 2.23 (3H, s): This singlet indicates 3 equivalent hydrogen atoms. These are likely part of a methyl group (CH₃) that is not split by neighboring hydrogens.
• δ 3.79 (3H, s): This singlet also indicates 3 equivalent hydrogen atoms, suggesting another methyl group (CH₃), possibly adjacent to an electronegative atom like oxygen (e.g., methoxy group -OCH₃).
• δ 7.04 (2H, ddd, J = 8.3, 1.3, 0.4 Hz): This complex splitting pattern (doublet of doublet of doublets) indicates 2 equivalent hydrogen atoms in an aromatic environment, with coupling constants suggesting interactions with neighboring hydrogens.
• δ 7.97 (2H, ddd, J = 8.3, 1.6, 0.4 Hz): Another complex splitting pattern, also indicating 2 equivalent hydrogen atoms in an aromatic environment, with similar coupling constants indicating interactions with neighboring hydrogens.
2. **Determine the Number of Different Types of Hydrogen Environments:**
• **Aliphatic Hydrogens:**
• 3 equivalent hydrogens at δ 2.23 (likely CH₃ group)
• 3 equivalent hydrogens at δ 3.79 (likely CH₃ group, possibly part of a methoxy group)
• **Aromatic Hydrogens:**
• 2 equivalent hydrogens at δ 7.04
• 2 equivalent hydrogens at δ 7.97
• The molecule has a total of 4 distinct types of hydrogen atoms:
• **Methyl group hydrogens:**
• 3 equivalent hydrogens at δ 2.23 (CH₃)
• 3 equivalent hydrogens at δ 3.79 (likely part of a methoxy group, -OCH₃)
• **Aromatic hydrogens:**
• 2 equivalent hydrogens at δ 7.04
• 2 equivalent hydrogens at δ 7.97

Figure 11: Human annotated C-NMR Elucidation

Based on the observation in spectrum interpretation, it is safe to say that the GPT-4o model has a basic grasp of the concept and the task at hand, but their deficiency mainly resides in obtaining fully correct information from the spectrum images and also they lack in-depth understanding. As for the problems in Stage 3, although the GPT-4o outputs are mostly correct, they are too general which shows the poor understanding of GPT-4o models in solving this intricate task. The stage 3 task can be roughly broken down into 3 subtasks: obtaining the correct information from the spectrum image, deducing the correct structural information from the spectral information, and finally translating this structural information into a correct molecular structure. GPT-4o models seem to perform well in the second subtask, and moderately for simple structures in the third subtask but seem to be especially struggling with the first subtask in the case of NMR spectra. This indicates the gap in current LLMs in fully interpreting data therefore more advanced models and approaches should be developed to tackle the problem.

### C.1.3 Complex Molecules

In addition to presenting molecules extracted from textbooks, we also demonstrate how the large language model (LLM) handles complex molecular structures. As illustrated in Figure 12, complex molecules typically have a larger pool of fragments. This expansion results in a greater number of valid elucidation paths, complicating the selection process for an appropriate starting point. Successfully navigating this enlarged pool necessitates an in-depth understanding of each fragment's properties and the associated, more intricate NMR data. In this context, LLMs may struggle because they often lack the nuanced chemical intuition and detailed analytical capabilities that human experts possess. Such limitations can lead to inaccuracies in interpreting complex interactions within NMR spectra, making LLMs less reliable.

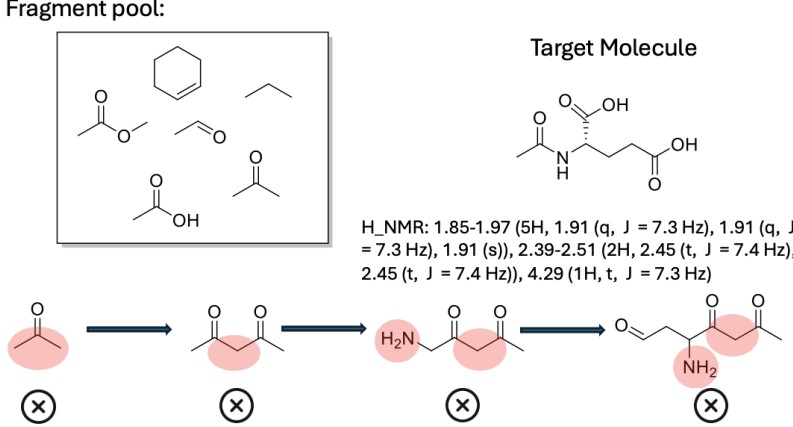

Figure 12: Complex molecule Structure Elucidation

## D    Compute Resources

For the execution of various models in our experiments, distinct compute resources were utilized based on the model's accessibility and computational requirements. Specifically, for models like Claude 3, GPT, and Gemini, we employed API calls to facilitate their operation, leveraging the existing infrastructure provided by their respective platforms. This approach allowed us to access these models without the need for local computational resources, thereby streamlining the process. Conversely, for all other open-sourced models employed in our study, we conducted the experiments locally using an NVIDIA A100 GPU. This high-performance computing unit was chosen due to its advanced capabilities in handling extensive computations and large model requirements efficiently.

