# OpenReview forum: "Can LLMs Solve Molecule Puzzles? A Multimodal Benchmark for Molecular Structure Elucidation"
_NeurIPS.cc/2024/Datasets_and_Benchmarks_Track — NeurIPS 2024 Track Datasets and Benchmarks Spotlight_

### Official Review · Reviewer_TXir · 2024-07-17
**MolPuzzle review**

**Rating:** 7
**Confidence:** 3
**Clarity:** See Clarity in Review

**Review:**

Quality

- Both the chemistry and computer science in the work are sound, which is nice to see.
- What about comparing to LLMs specifically trained on scientific data like Galatica?
- Have you tried presenting the spectra to an LLM in the form of language, e.g. “There is a peak at 2 ppm”? Could better indicate whether there are issues with the vision model in detecting fine-grained peaks.
- “MACCS FTS of 0.431 suggests that the generated molecules maintain a significant level of structural similarity.” How do you know it’s significant, is there some sort of baseline you can include to convince readers it’s significant? Also, can you provide some examples of elucidated molecules in appendix compared to ground truth?

Clarity

-  Should provide formula for degree of saturation (and also definition) — may not be clear to computer science audience. Do you provide formula when prompting llms?
- In Figure 3, there are 234 molecules but only 176 have molecular formula questions in Stage 1. This seems like a simple task to generate a QA pair for, why are there some missing molecules?
- Stage 2 — for spectra query templates like “What does the ion peak in 3200-3600 suggest?”, do you provide the option for saying “nothing?” if there isn’t one? It seems like you’re phrasing the question in a way that assumes there is a peak, but what if there isn’t one? It would be helpful to provide a list of all the templates used for the three stages, not just samples
- The final stage — the LLM is instructed to iteratively construct a molecule. can you provide more details about this? What is the maximum number of iterations and what if some identified fragments are wrong or missing, is the model allowed to e.g. restart or change its pool of fragments if it can’t successfully build on first attempt?
- 4.1.2 isnt clear — did you have the llm try to deduce the molecular structure in a single step instead of breaking it down into three stages? How did you provide the input information?
- Figure 5 caption should describe what the errors are
- In A.1, you write that “For additional spectra that were not available, we used simulation methods”. How accurate are these simulation methods?
- In A.1 you say that you had to remove 34 molecules due to copyright concerns, are these used anywhere or published online in your repo? Else I would report 200 molecules since that is what users would have access to. Also, how does this number compare to other dataset sizes

Originality

- The work is original to my knowledge; I am not aware of a dataset for molecule structure elucidation using a similar puzzle setup as in chemistry exams/de novo structure characterization.

Significance

- Structural elucidation is an important problem in chemistry and current models usually don’t approach the problem from a multimodal angle (usually models handle only one spectrum at a time); however, chemists would normally consider multiple spectra at once (unless they are looking at specific changes to a known molecular scaffold, for example) and this is how students are trained in chemistry courses. Thus, I would say a benchmark like this is quite interesting.

**Strengths:**

- The work showcases molecular structure elucidation using multiple modalities which is more similar to how it would be done in real life (as a puzzle, which is similar to how chemists would view the task); we move towards ML models being able to analyze several spectra at once, and so it’s nice that this work highlights that
- The work is well-written
- It seems like there are a lot of QA pairs but only a few examples provided in paper (can’t access dataset link). From the examples it seems like the questions mimic the questions a chemistry student (or chemist) would ask when looking at the spectra

**Additional Feedback:**

- “we developed 26 unique templates (see Appendix A.1 for details)” —> seems to be A2 and A3? Please check that appendix numbers
- “open-sourced models like LLama3 ” — you only tested one open-source model, I wouldn’t make this generalization to all open-sourced models

**Correctness:**

- To my knowledge, the claims in the submission are correct.
- I would like some clarification on how the QA pairs for Stage 3 are created, I had to re-read lines 210-216 a few times and I still feel like I don’t fully understand.

**Documentation:**

The provided URL leads to a 404 error! (Noted on July 17 2024)

**Ethics:**

- Unsure if human involvement requires IRB number (number not provided in appendix).

**Limitations:**

- Authors addressed limitations of existing methods
- Authors write that negative societal impacts are addressed in Section 6 but I don’t see it

**Opportunities For Improvement:**

- For Stage 2 did you try writing out the spectrum instead of just showing the image to isolate whether mistakes being made are because of visual processing or reasoning?
- The flow of how everything is a bit confusing — how were Stage 3 QAs created? How did you prompt the LLM to solve the puzzle in one go instead of breaking it down into three stages (4.1.2) and what are the main takeaways from those results?
- Appendix should have all templates

**Relation To Prior Work:**

Only provided 2 references to structure elucidation but there are others, such as:
[1] https://arxiv.org/abs/2404.02360
[2] https://openreview.net/forum?id=4ilKwquW51

**Summary And Contributions:**

Authors propose MolPuzzle: a QA dataset that combines multiple spectra to test models’ abilities to elucidate molecular structures in a way that is more similar to how chemists and chemistry students approach the task. Authors show that existing models are quite far from human performance, but out of existing models, GPT-4o does the best.

---

> ### Author Rebuttal · Authors · 2024-08-16
>
> We thank the reviewer for their positive feedback and detailed review. Please find our responses to your questions and concerns below. We will incorporate the useful discussion and additional experiments in the updated version of our paper. We would also appreciate it if you could consider raising the score based on our revisions and clarifications.
>
> `Q1: What about comparing to LLMs specifically trained on scientific data like Galatica?`
>
> Thank you for the advice. We have expanded the scope of our work to include Galactica-30b in our experiments for stage 1, which will be reflected in the revised paper.
>
> >| Method        | SI                   | ARI                | FGI                | SDC  |
> >|---------------|----------------------|--------------------|--------------------|------|
> >| Galactica-30b| $0.826 \pm 0.248$     | $0.347 \pm 0$      | $0.46 \pm 0.005$   | 0    |
> >
> >   Table 1: F1 score of Galactica-30b in Stage-1 (Molecule Understanding) Tasks.
>
>
> `Q2: Have you tried presenting the spectra to an LLM in the form of language, e.g. “There is a peak at 2 ppm”? Could better indicate whether there are issues with the vision model in detecting fine-grained peaks.`
>
> Thank you for your valuable suggestion. You are correct that using a more descriptive format, such as "There is a peak at 2 ppm," could clarify input for the model. However, most available chemistry data is presented in a standard format, which includes not only peak shifts but also additional information. For example, multiple spectrum processing tools can convert NMR spectrum images into text format, e.g. ¹H NMR (400 MHz, CDCl₃): δ 1.10 (t, 3H, J = 7.0 Hz, CH₃), 3.65 (q, 2H, J = 7.0 Hz, CH₂), 4.70 (s, 1H, OH). It is challening to fully understand the peak shifts and other important patterns from such descriptive text.  We are currently investigating how effectively LLMs can capture molecular structural information from these standard NMR text representations. We will include a case study in our revised version to further explore this idea.
>
> `Q3: “MACCS FTS of 0.431 suggests that the generated molecules maintain a significant level of structural similarity.”`
>
> We used the term "significant level of structural similarity" to base the benchmark similarity achieved. In the revised version, we will refine the wording to improve clarity.
>
>
>
> `Q4: Should provide formula for degree of saturation (and also definition) — may not be clear to computer science audience. Do you provide formula when prompting llms?`
>
> We appreciate your thoughtful advice and will revise our paper by providing detailed descriptions of all the chemistry concepts involved. Regarding the question, “Do you provide formula when prompting llms?” the answer is yes. We explicitly include the molecular formula in our prompts, as our goal is to test the LLMs’ capability to deduce molecular substructures based on the corresponding formula.
>
> `Q5: In Figure 3, there are 234 molecules but only 176 have molecular formula questions in Stage 1. This seems like a simple task to generate a QA pair for, why are there some missing molecules?`
>
> Apologize for the confusion, it is common for molecules to have different structure but same molecular formula. Therefore, the number of molecules is different from that invovled in questions. We add this clarification to reduce misunderstanding.
>
> `Q6: Stage 2 — for spectra query templates like “What does the ion peak in 3200-3600 suggest?”, do you provide the option for saying “nothing?” if there isn’t one? It seems like you’re phrasing the question in a way that assumes there is a peak, but what if there isn’t one? It would be helpful to provide a list of all the templates used for the three stages, not just samples`
>
> We apologize for the confusion. In Stage 2, we have designed question templates that cover a broad range of substructure-related tasks, including peak identification, functional group identification, and more. The spectrum image like IR usually has multiple peaks. We ensure that questions are phrased to accommodate cases where there may be no relevant peak. For a comprehensive overview, the details of all the question templates used across the three stages are available in our GitHub repository.

---

> > ### Author Rebuttal · Authors · 2024-08-16
> >
> > We thank the reviewer for their positive feedback and detailed review. Please find our responses to your questions and concerns below. We will incorporate the useful discussion and additional experiments in the updated version of our paper. We would also appreciate it if you could consider raising the score based on our revisions and clarifications.
> >
> > `Q1: What about comparing to LLMs specifically trained on scientific data like Galatica?`
> >
> > Thank you for the advice. We have expanded the scope of our work to include Galactica-30b in our experiments for stage 1, which will be reflected in the revised paper.
> >
> > >| Method        | SI                   | ARI                | FGI                | SDC  |
> > >|---------------|----------------------|--------------------|--------------------|------|
> > >| Galactica-30b  | $0.826 \pm 0.248$     | $0.347 \pm 0$      | $0.46 \pm 0.005$   | 0    |
> > >
> > >   Table 1: F1 score of Galactica-30b in Stage-1 (Molecule Understanding) Tasks.
> >
> >
> > `Q2: Have you tried presenting the spectra to an LLM in the form of language, e.g. “There is a peak at 2 ppm”? Could better indicate whether there are issues with the vision model in detecting fine-grained peaks.`
> >
> > Thank you for your valuable suggestion. You are correct that using a more descriptive format, such as "There is a peak at 2 ppm," could clarify input for the model. However, most available chemistry data is presented in a standard format, which includes not only peak shifts but also additional information. For example, multiple spectrum processing tools can convert NMR spectrum images into text format, e.g. ¹H NMR (400 MHz, CDCl₃): δ 1.10 (t, 3H, J = 7.0 Hz, CH₃), 3.65 (q, 2H, J = 7.0 Hz, CH₂), 4.70 (s, 1H, OH). It is challenging to fully understand the peak shifts and other important patterns from such descriptive text.  We are currently investigating how effectively LLMs can capture molecular structural information from these standard NMR text representations. We will include a case study in our revised version to further explore this idea.
> >
> > `Q3: “MACCS FTS of 0.431 suggests that the generated molecules maintain a significant level of structural similarity.”`
> >
> > We used the term "significant level of structural similarity" to base the benchmark similarity achieved. In the revised version, we will refine the wording to improve clarity.
> >
> >
> >
> > `Q4: Should provide formula for degree of saturation (and also definition) — may not be clear to computer science audience. Do you provide formula when prompting llms?`
> >
> > We appreciate your thoughtful advice and will revise our paper by providing detailed descriptions of all the chemistry concepts involved. Regarding the question, “Do you provide formula when prompting llms?” the answer is yes. We explicitly include the molecular formula in our prompts, as our goal is to test the LLMs’ capability to deduce molecular substructures based on the corresponding formula.
> >
> > `Q5: In Figure 3, there are 234 molecules but only 176 have molecular formula questions in Stage 1. This seems like a simple task to generate a QA pair for, why are there some missing molecules?`
> >
> > Apologize for the confusion, it is common for molecules to have different structures but the same molecular formula. Therefore, the number of molecules is different from that involved in questions. We add this clarification to reduce misunderstanding.
> >
> > `Q6: Stage 2 — for spectra query templates like “What does the ion peak in 3200-3600 suggest?”, do you provide the option for saying “nothing?” if there isn’t one? It seems like you’re phrasing the question in a way that assumes there is a peak, but what if there isn’t one? It would be helpful to provide a list of all the templates used for the three stages, not just samples`
> >
> > We apologize for the confusion. In Stage 2, we have designed question templates that cover a broad range of substructure-related tasks, including peak identification, functional group identification, and more. The spectrum image like IR usually has multiple peaks. We ensure that questions are phrased to accommodate cases where there may be no relevant peak. For a comprehensive overview, the details of all the question templates used across the three stages are available in our GitHub repository.

---

> > > ### Author Response · Authors · 2024-08-16
> > > **Following rebuttal**
> > >
> > > `Q7: The final stage — the LLM is instructed to iteratively construct a molecule. can you provide more details about this? What is the maximum number of iterations and what if some identified fragments are wrong or missing, is the model allowed to e.g. restart or change its pool of fragments if it can’t successfully build on first attempt?`
> > >
> > > We apologize for the confusion. In the final stage, we utilize a valid fragment pool gathered from Stages 1 and 2 for substructure selection and assembly. The process involves the following steps:
> > >
> > > 	1.	Initial Fragment Selection: We prompt the LLMs with the fragment pool (in the form of SMILES string) and the NMR description to select an initial fragment for assembly. The selected fragment is extracted and stored for further processing.
> > > 	2.	Chemical Formula and Unsaturation Check: The selected fragment is converted into its chemical formula and unsaturation degree. These values are then subtracted from the original molecular formula and degree of unsaturation to determine the remaining atoms and unsaturation.
> > > 	3.	Iterative Fragment Assembly: If the remaining molecular formula has multiple main atoms, we prompt the LLM to select additional fragments from the pool, considering the remaining atoms and unsaturation. The selected fragments are then concatenated to form a potential molecule.
> > > 	4.	Molecule Validation and NMR Matching: The new molecule is validated using RDKit to check if the fragments can be connected. If multiple valid molecules are possible, the LLM ranks them based on their NMR match. The molecule that best matches the NMR data is selected for further assembly.
> > > 	5.	Subsequent Assembly and Adjustment: The process of fragment selection, validation, and molecule assembly continues iteratively. After each successful connection, the new molecule’s chemical formula is compared against the original formula, and the remaining atoms and unsaturation are recalculated.
> > > 	6.	Termination Conditions: The assembly process is terminated if no valid fragments can be selected, if the molecular formula is fully satisfied, or if the maximum of 5 iterations is reached.
> > > 	7.	Final Output: The final assembled molecule, along with the intermediate stages, is recorded. If the process successfully yields a molecule that fits the original molecular formula and NMR data, it is considered a valid solution.
> > >
> > >
> > > `Q8: 4.1.2 isnt clear — did you have the llm try to deduce the molecular structure in a single step instead of breaking it down into three stages? How did you provide the input information?`
> > >
> > > Apology for the confusion caused.  As mentioned in the previous answer, we break down the tasks into three stages. The first two stages involve gathering possible substructures based on the molecular formula and spectrum images. In these stages, the inputs include text prompts, spectrum images, and the molecular formula. The final stage involves assembling the substructures. For this stage, the inputs are text and the substructures in the form of SMILES string.
> > >
> > > `Q9: Figure 5 caption should describe what the errors are`
> > >
> > > Thank you for your thoughtful advice, the updated caption is: Mismatch in hydrogen NMR shifts.
> > > We will also update it in our revised paper.
> > >
> > >
> > > `Q10: In A.1, you write that “For additional spectra that were not available, we used simulation methods”. How accurate are these simulation methods?`
> > >
> > > The mean absolute errors for simulated NMR data range from 0.6 to 0.8 Hz, which present minimal concerns for accurately representing molecular structures and do not impact molecular construction tasks.

---

> > > > ### Author Response · Authors · 2024-08-16
> > > > **following rebuttal**
> > > >
> > > > `Q11: In A.1 you say that you had to remove 34 molecules due to copyright concerns, are these used anywhere or published online in your repo? Else I would report 200 molecules since that is what users would have access to. Also, how does this number compare to other dataset sizes`
> > > >
> > > > The 34 molecules were removed from the dataset due to copyright concerns, and they are not included or published in our repository. However, we are actively working on generating alternative molecules to replace them, with the goal of making the complete set available to users. For now, users will have access to the 200 molecules, and we will update the manuscript to reflect this.
> > > >
> > > > `Q12: Authors write that negative societal impacts are addressed in Section 6 but I don’t see it`
> > > >
> > > > We are sorry for the oversight, here is our discussion regarding negative societal impacts and we will also update it in our revised manuscript:
> > > >
> > > > > Automating molecular elucidation using LLMs has significant benefits but also poses serious risks, especially regarding the creation of prohibited drugs. 1.)Facilitation of Illicit Drug Synthesis: LLMs could be used to design new synthetic drugs that evade current regulations, making it easier for illicit manufacturers to produce harmful substances. 2.)Lowering the Barrier to Entry: The technology could enable individuals with minimal expertise to create detailed molecular blueprints for prohibited drugs, increasing the potential for misuse. 3.)	Regulatory Challenges: The rapid generation of novel compounds could overwhelm drug regulators, leading to delays in banning new synthetic drugs and complicating the control of harmful substances. 4.)	Ethical and Legal Issues: Questions about responsibility and access to such powerful tools arise. Regulating who can use these technologies and for what purposes becomes crucial to prevent misuse.
> > > >
> > > >
> > > >
> > > > `Q13: Unsure if human involvement requires IRB number (number not provided in appendix).`
> > > >
> > > >
> > > > Thank you for pointing this out. We consulted the IRB approval team at our university, and they provided the following response: Since the questions do not pertain to individuals and no personal data is being collected, this protocol falls under an exempt category of IRB review.
> > > >
> > > >
> > > > `Q14: Only provided 2 references to structure elucidation but there are others, such as: [1] https://arxiv.org/abs/2404.02360 [2] https://openreview.net/forum?id=4ilKwquW51`
> > > >
> > > > Thank you for your advice. We will include these papers in our related work session.
> > > >
> > > > `Q15: “open-sourced models like LLama3 ” — you only tested one open-source model, I wouldn’t make this generalization to all open-sourced models`
> > > >
> > > > Thank you for your suggestion. While it’s true that we only tested LLama 3, which is currently recognized as one of the best open-source LLMs in the community, we focused on this model due to its strong performance and wide adoption. However, we acknowledge that our findings may not necessarily generalize to all open-source models.
> > > >
> > > > [1] Binev, Yuri, Maria MB Marques, and João Aires-de-Sousa. "Prediction of 1H NMR coupling constants with associative neural networks trained for chemical shifts." Journal of chemical information and modeling 47.6 (2007): 2089-2097.

---

> > > > > ### Author Response · Authors · 2024-08-27
> > > > > **Appreciation for Your Feedback**
> > > > >
> > > > > We greatly value your insightful and comprehensive feedback.
> > > > >
> > > > > As the discussion period is nearing its end, we kindly remind you to review our response. We are eager to receive your thoughts.
> > > > >
> > > > > Thank you again for your time and attention.
> > > > >
> > > > > Warm regards,
> > > > >
> > > > > The Authors

---

> > > > > > ### Author Response · Authors · 2024-08-28
> > > > > >
> > > > > > Thank you for your thoughtful review and feedback, we truly appreciate your time and expertise.

---

> > > > > > > ### Comment · Reviewer_TXir · 2024-08-30
> > > > > > >
> > > > > > > Thank you to the authors for answering my questions and performing additional experiments. I have raised my score.

---

> > > > > > > > ### Author Response · Authors · 2024-08-30
> > > > > > > > **Thank you!**
> > > > > > > >
> > > > > > > > Thank you for taking the time to reconsider our work. We truly appreciate your support and are glad that our additional efforts were helpful.

---

### Official Review · Reviewer_hyGW · 2024-07-18
**MolPuzzle, new benchmark for LLM**

**Rating:** 8
**Confidence:** 4
**Correctness:** Crrect.
**Clarity:** Yes, it's clear.

**Review:**

Pros
1. MolPuzzle provides a novel and comprehensive benchmark for evaluating LLMs in molecular structure elucidation.
2. The paper presents an extensive evaluation of multiple state-of-the-art LLMs, offering valuable insights into their capabilities and limitations.
3. The work bridges AI and chemistry, ensuring the benchmark is both scientifically relevant and technically robust.
4. MolPuzzle includes a wide range of tasks and varying difficulty levels, facilitating thorough evaluations of LLMs.
5. This benchmark probvided a great way to evaluate the LLM performance on the molecule and biology by capture the spectra information to help reconstruct the molecules.

Cons
1. The LLMs demonstrate limited understanding and reasoning capabilities in complex chemistry concepts, which are essential for molecule construction and interpretation of spectral data.
2. The dataset may not cover the full spectrum of molecular diversity encountered in practical applications. Limited diversity can result in models that perform well on the benchmark but fail to generalize to real-world scenarios. It would be great if authors could discuss the diversity of the dataset.

**Strengths:**

See Review.

**Additional Feedback:**

Please check the url link share in the paper.

**Documentation:**

The URL cannot be visit.

**Limitations:**

The authors indicated the limitations.

**Opportunities For Improvement:**

The benchmark could be extended with more data diversity, include different kiind of structures/biology class for the molecules.

**Relation To Prior Work:**

Yes.

**Summary And Contributions:**

The authors proposed MolPuzzle,  a new benchmark designed to evaluate the capabilities of Large Language Models (LLMs) in solving molecular structure elucidation tasks. This complex problem involves deducing a molecule's structure from various types of spectral data, akin to solving a crossword puzzle. The MolPuzzle dataset consists of 234 instances of structure elucidation, featuring over 18,000 QA samples across three interlinked sub-tasks: molecule understanding, spectrum interpretation, and molecule construction. The authors evaluate the performance of over 10 LLMs, including GPT-4o, Gemini-pro, and Claude-3-opus.

---

> ### Author Rebuttal · Authors · 2024-08-16
>
> We thank the reviewer for their positive feedback and detailed review. Please find our responses to your questions and concerns below. We will add discussion to the related work.
>
> `Q1: The LLMs demonstrate limited understanding and reasoning capabilities in complex chemistry concepts, which are essential for molecule construction and interpretation of spectral data. `
>
> We agree that while current LLMs show less promise in understanding complex chemistry concepts, there is still room for improvement, especially in performing multi-step reasoning and planning tasks essential for molecule construction and interpreting spectral data. To address these limitations, one potential direction is to fine-tune multimodal LLMs with spectrum images and domain-specific knowledge, which could enhance their understanding of molecular structures and spectral interpretation. Additionally, providing LLMs with feedback at each iteration during substructure selection and assembly could further improve their accuracy and interpretability.
>
>
> `Q2: The dataset may not cover the full spectrum of molecular diversity encountered in practical applications. Limited diversity can result in models that perform well on the benchmark but fail to generalize to real-world scenarios. It would be great if authors could discuss the diversity of the dataset.`
>
> We thank the reviewer for their insightful suggestion. The molecules included in our benchmark are sourced from a graduate-level textbook, representing a diverse array of sizes, shapes, and complexities. This diversity is reflected in the variations in bonding patterns and functional groups among the molecules. In the revised version, we will include a 2D molecular space map to better illustrate this diversity.
>
>
> `Q3: The URL cannot be visit.`
>
> We apologize for the oversight. The GitHub link has now been made accessible.

---

> > ### Comment · Reviewer_hyGW · 2024-08-27
> >
> > Thanks author for the detailed response. I will keep my accepted score.

---

> > > ### Author Response · Authors · 2024-08-30
> > > **Thank you!**
> > >
> > > Thank you for your insightful feedback and thoughtful review. We genuinely appreciate the time and expertise you’ve dedicated to this.

---

### Official Review · Reviewer_bdDQ · 2024-07-25
**Nice paper presenting novel multimodal benchmark for determination of molecular structure**

**Rating:** 7
**Confidence:** 4

**Review:**

The paper presents a novel benchmark, MolPuzzle, for the task of molecular structure elucidation using LLMs. The benchmark is multi-modal, requiring understanding of both text (instructions) and images (graphs/spectra) to solve the problems in the benchmark.

**Quality:** The idea for the paper is interesting, as I haven’t seen any benchmarks like this one for molecular structure elucidation, particularly the multi-modal aspect (feeding images to LLMs). Overall, an good idea, interesting benchmark, executed well. However, the link to the GitHub repository doesn’t work (404), so I was unable to inspect the code.

**Clarity:** The paper is well-written and follows the standard organisation of a paper, making it easy to follow. The figures, particularly Figure 2, was helpful to understand the input/output of the tasks.

**Originality:** I haven’t seen a benchmark for molecular structure elicidation in the literature, and would agree with the authors that it is novel.

**Significance:**
-	To the AI community: Nice contribution, highlighting that complex tasks in the sciences are still underexplored
-	To the chemistry community: Not that significant, since the LLMs are not able to actually solve the problem of structure elucidation
-	To the AI for chemistry community: Excellent contribution, a new frontier of multi-modal tasks that LLMs are not yet proficient in solving, I look forward to seeing future papers written to improve on the performance seen in this paper.

**Pros:**
- Introduction of a novel and challenging benchmark for AI in chemistry.
- Comprehensive dataset with well-defined sub-tasks.
- Rigorous evaluation of multiple state-of-the-art LLMs.
- Great that LLM performance was compared to the performance of human experts.

**Cons:**
- Performance of the LLMs on tasks 2 and 3 was low, with limited discussion on how the performance could be improved.
- No working link to the code repository/data.

**Strengths:**

See above.

-	Social implications: The benchmark has the potential to accelerate the development of (LLM) models that can perform complex scientific tasks, specifically molecular structure elucidation. Any work that accelerates and automates the work that chemists do today has the potential to accelerate discoveries & developments in the chemical industry and health care/drug discovery.
-	Ethical implications: N/A

**Additional Feedback:**

Overall, a nice paper, I look forward to seeing what researchers do with this dataset in the future!

However, the data & code should be made available before making a decision!

**Clarity:**

The paper is well-written, with a clear and logical structure. The figures are good, and particularly Figure 2 helps with understanding the paper.

**Correctness:**

The claims made in the paper appear reasonable and correct, however, it’s hard to verify without access to the code/data.

**Documentation:**

The link to the code/data doesn’t link to anything (404 error), so I was unable to inspect the documentation.

**Ethics:**

The paper does not raise any ethical concerns.

**Limitations:**

The authors present an interesting and useful benchmark, which was thoroughly evaluated on different LLM models, and compared to an expert chemist baseline. However, they do not explore (or really even mention) fine-tuning of models to improve performance on their benchmark dataset. The paper would have been more impressive had the authors presented a fine-tuned LLM model which outperformed GPT-4o. Fine-tuning models may be beyond the scope of this paper, so I look forward to reading about a fine-tuned LLM model in a future paper.

**Opportunities For Improvement:**

1.	Provide a working link to the code/data.
2.	The related work section is incomplete: attempting to determine molecular structures in an automated fashion is not a new idea, and I would like to see other approaches mentioned (e.g. rule-based or other ML approaches). How well do these approaches work. I’m not sure if it’s too much work to benchmark existing methods on your benchmark dataset, but they should definitely be mentioned in the related work section.
3.	The authors found that LLMs do not perform well on tasks 2 and 3 but stop short of suggesting areas of future research to improve the performance. They mention “significant opportunity to develop specialized multimodal LLMs” but stop short in suggesting how such a “specialized multimodal LLMs” would actually be built (fine-tuning?). It would be great if the authors expanded on this.
4.	I don’t understand the “Answer” to 1. In Figure 2. I expected the “Answer” would be a list of all possible substructures with that chemical formula (e.g. [C, CC, CO, …]), and not just Carboxylic acid (yes/no)?
5.	Minor, but please change “way worse” -> “much worse” (more formal)

**Relation To Prior Work:**

I agree with the authors that using LLMs for molecular structure elucidation is underexplored, however, they could do more to explain how their work stands in relation to prior ML approaches for analysis of spectra.

**Summary And Contributions:**

This paper introduces MolPuzzle, a new benchmark designed to test the capabilities of Large Language Models (LLMs) in molecular structure elucidation. This task involves deducing a molecule’s structure from various data sources, notably including various types of spectral data. The benchmark consists of 234 molecular structures, featuring over 18,000 Q&A samples across three interlinked sub-tasks: molecule understanding, spectrum interpretation, and molecule construction. The authors evaluate 10 LLMs and find that while GPT-4o performs best among LLM models (and performs near perfectly on task 1), the LLMs perform significantly worse than humans on tasks 2 and 3. This work highlights that LLMs still have a way to go to compete with humans on complex scientific tasks and presents a novel benchmark on molecular structure elucidation to the community.

---

> ### Author Rebuttal · Authors · 2024-08-16
>
> We thank the reviewer for their positive feedback and detailed review. Please find our responses to your questions and concerns below, which we will also implement together within the updated version of the paper.
>
> `Q1: Performance of the LLMs on tasks 2 and 3 was low, with limited discussion on how the performance could be improved. Suggesting areas of future research to improve the performance.`
>
> For Task 2, which involves interpreting spectral images, we identified a key limitation: most multimodal LLMs (including GPT-4 and Claude-3) are not well-aligned with spectral data. This misalignment limits their ability to accurately identify finer details. For instance, while these models can effectively identify and analyze large spikes in the spectral images, their performance significantly deteriorates when dealing with smaller spikes or unusual waveforms. A promising future direction would be to improve the alignment of LLMs with spectral data, which could enhance performance in Task 2.
>
> For Task 3, which involves multi-step planning and reasoning within LLMs, one potential improvement is to provide feedback at each step, allowing the model to correct its selections as it progresses. Another approach could involve using Retrieval-Augmented Generation (RAG) and fine-tuning techniques, where additional relevant information is injected to guide the LLMs toward making more accurate decisions.
>
> We thank the reviewer for their insightful advice and will update our manuscript accordingly.
>
> `The related work section is incomplete: attempting to determine molecular structures in an automated fashion is not a new idea, and I would like to see other approaches mentioned (e.g. rule-based or other ML approaches). .`
> We thank the reviewer for the insightful suggestion. As you mentioned, there are two categories of research work for automating molecular structure determination: rule-based and machine-learning approaches. The rule-based methods [1, 2] primarily rely on predefined rules and heuristics derived from known chemical principles and spectral patterns. These methods systematically apply these pre-defined rules to interpret spectral data, identify functional groups, and deduce possible molecular structures. While rule-based approaches can be highly effective for well-characterized molecules and standard spectral features, they often struggle with complex or novel compounds where the spectral patterns may not conform to the established rules. Additionally, many rule-based methods are embedded in commercial software that is not publicly accessible for free, making it challenging to use them for benchmarking studies [2].
>
> On the other hand, machine learning approaches [3,4] leverage large datasets of spectral and structural information to train models that can predict molecular structures from spectral data. These models can learn complex patterns and correlations that may not be captured by rule-based methods, making them more adaptable to a wide range of molecules, including those that are difficult to analyze using traditional techniques. However, machine learning approaches require substantial amounts of high-quality training data and may require extensive computational resources. Additionally, the interpretability of machine learning models limits their application, as they may not provide clear, rule-based explanations for their predictions.
>
> **Regarding the benchmarking** of existing methods on our dataset, we acknowledge the importance of such an evaluation. While it may be beyond the scope of this paper to perform a detailed benchmarking of all existing methods, we agree that a thorough discussion of these methods, along with their strengths and limitations, is essential for providing context to our work. We thus will ensure that the related work section is expanded to include a more comprehensive review of both rule-based and ML approaches. This will also help our followers develop more advanced AI models and benchmark their work.
>
> [1]. Chylek, Lily A., et al. "Rule‐based modeling: a computational approach for studying biomolecular site dynamics in cell signaling systems."
> [2].Tsugawa, Hiroshi, et al. "Hydrogen rearrangement rules: computational MS/MS fragmentation and structure elucidation using MS-FINDER software." Analytical chemistry 88.16 (2016): 7946-7958.
> [3]. Young, Adamo, et al. "FraGNNet: A Deep Probabilistic Model for Mass Spectrum Prediction." arXiv preprint arXiv:2404.02360 (2024).
> [4]. Schilter, Oliver, et al. "Unveiling the Secrets of $^ 1$ H-NMR Spectroscopy: A Novel Approach Utilizing Attention Mechanisms." AI for Accelerated Materials Design-NeurIPS 2023 Workshop. 2023.
>
>
> `Q3: I don’t understand the “Answer” to 1. In Figure 2. I expected the “Answer” would be a list of all possible substructures with that chemical formula (e.g. [C, CC, CO, …]), and not just Carboxylic acid (yes/no)?`
>
>
> Thank you for your question and for highlighting this potential confusion. We understand that you expected the “Answer” to include a list of all possible substructures corresponding to a given chemical formula. However, in the context of our work, the “Answer” in Figure 2 is designed to assess how well LLMs understand specific patterns of a molecule. For example, the question about carboxylic acid presence is intended to test the model’s ability to recognize this particular functional group.
>
> In addition to this, we have included other questions to evaluate different molecular characteristics, such as the “degree of saturation”,  “presence of aromatic rings.”, etc.  All these templates and questions have been made available in our GitHub repository for further reference and transparency.
>
> `Q4: Minor, but please change “way worse” -> “much worse” (more formal)`
>
> Thanks for the advice, we will revise our paper accordingly.

---

> > ### Author Response · Authors · 2024-08-16
> > **following rebuttal**
> >
> > `Q5: The authors present an interesting and useful benchmark, which was thoroughly evaluated on different LLM models, and compared to an expert chemist baseline. However, they do not explore (or really even mention) fine-tuning of models to improve performance on their benchmark dataset. The paper would have been more impressive had the authors presented a fine-tuned LLM model which outperformed GPT-4o. Fine-tuning models may be beyond the scope of this paper, so I look forward to reading about a fine-tuned LLM model in a future paper.`
> >
> > Thank you for your constructive feedback, which we appreciate greatly. We agree that fine-tuning would improve the performance of current LLMs, and this is also our on-going work. We are actively working on developing a fine-tuned LLM model, and we look forward to sharing our findings in a future publication.

---

> > > ### Comment · Reviewer_bdDQ · 2024-08-18
> > >
> > > Thank you for incorporating my feedback, and for the clarifications, these are good changes!

---

> > > > ### Author Response · Authors · 2024-08-18
> > > > **Thank you**
> > > >
> > > > Thank you for your thoughtful review and feedback, we truly appreciate your time and expertise.

---

### Author Rebuttal · Authors · 2024-08-16

Dear reviewers and AC,

We would like to thank the reviewers for their insightful feedback, which has provided us with excellent guidance towards crafting a more comprehensive and competitive paper! As highlighted by the reviewers, our work evaluates the capabilities of Large Language Models (LLMs) in solving molecular structure elucidation tasks. We are encouraged that the reviewers find our benchmark “novel and comprehensive,” “offering valuable insights” on “a new frontier of multi-modal tasks,” and “well-written” while effectively “bridging AI and chemistry.”

We are committed to sincerely addressing all of the reviewers’ concerns and have provided detailed responses to each of your comments below. In light of the feedback, we believe the enhancements we’ve made will resonate even more profoundly with our audience and meet an even higher standard, positioning our work to inspire a broader readership.

Additionally, we apologize for not making our GitHub repository public earlier. It is now accessible at this [link](https://github.com/KehanGuo2/MolPuzzle).


Thank you very much.

with best Regards,

Authors

---

### Decision · Program_Chairs · 2024-09-26

**Decision:**

Accept (Spotlight)

**Comment:**

In this manuscript, the authors propose MolPuzzle, a benchmark for evaluating LLMs' ability in molecule understanding, spectrum interpretation, and molecule construction.

Strengths:
1. The reviewers praised the novelty of this work.
2. The reviewers found this work technically sound.
3. The reviewers found the evaluations performed informative. The three different sub-tasks defined and the big difference in performance of the LLMs in these sub-tasks are insightful.

Weaknesses:
1. Some details were missing in the original version.
2. Since the LLMs did not perform well in sub-tasks 2 and 3, a reviewer suggested that the contribution of this work to chemistry is limited -- but this does not affect the value of this work in benchmarking the LLMs.
3. The GitHub repository was originally not accessible.

The authors and reviewers participated in post-review discussions actively and all reviewers were satisfied with the additional work done and revisions made by the authors. Two future directions were identified:
1. Testing if fine-tuning can improve the performance of the LLMs
2. Testing whether other ways to represent the spectral data, such as descriptions in a natural language, can improve the performance of the LLMs